# NKCC1 and KCC2 Chloride Transporters Have Different Membrane Dynamics on the Surface of Hippocampal Neurons

**DOI:** 10.3390/cells12192363

**Published:** 2023-09-26

**Authors:** Erwan Pol, Etienne Côme, Zaha Merlaud, Juliette Gouhier, Marion Russeau, Sophie Scotto-Lomassese, Imane Moutkine, Xavier Marques, Sabine Lévi

**Affiliations:** Institut du Fer à Moulin, Institut National de la Santé Et de la Recherche Médicale (INSERM) UMR-S 1270, Sorbonne Université, 75005 Paris, France; erwan.pol@inserm.fr (E.P.); comeetienne@gmail.com (E.C.); zaha.merlaud@inserm.fr (Z.M.); juliette.gouhier@sfr.fr (J.G.); marion.russeau@inserm.fr (M.R.); sophie.scotto@inserm.fr (S.S.-L.); imane.moutkine@inserm.fr (I.M.); xavier.marques@mnhn.fr (X.M.)

**Keywords:** hippocampus, chloride homeostasis, GABAergic transmission, single-particle tracking, quantum dot, STORM, membrane trafficking, synapses

## Abstract

Na-K-2Cl cotransporter 1 (NKCC1) regulates chloride influx in neurons and thereby GABA_A_ receptor activity in normal and pathological conditions. Here, we characterized in hippocampal neurons the membrane expression, distribution and dynamics of exogenous NKCC1a and NKCC1b isoforms and compared them to those of the chloride extruder K-Cl cotransporter 2 (KCC2). We found that NKCC1a and NKCC1b behave quite similarly. NKCC1a/1b but not KCC2 are present along the axon initial segment where they are confined. Moreover, NKCC1a/1b are detected in the somato-dendritic compartment at a lower level than KCC2, where they form fewer, smaller and less compact clusters at perisynaptic and extrasynaptic sites. Interestingly, ~60% of dendritic clusters of NKCC1a/1b are colocalized with KCC2. They are larger and brighter than those devoid of KCC2, suggesting a particular NKCC1a/1b-KCC2 relationship. In agreement with the reduced dendritic clustering of NKCC1a/1b compared with that of KCC2, NKCC1a/1b are more mobile on the dendrite than KCC2, suggesting weaker cytoskeletal interaction. NKCC1a/b are confined to endocytic zones, where they spend more time than KCC2. However, they spend less time in these compartments than at the synapses, suggesting that they can rapidly leave endocytic zones to increase the membrane pool, which can happen in pathological conditions. Thus, NKCC1a/b have different membrane dynamics and clustering from KCC2, which helps to explain their low level in the neuronal membrane, while allowing a rapid increase in the membrane pool under pathological conditions.

## 1. Introduction

Synaptic inhibition in the somato-dendritic compartment is mediated by chloride-permeable GABA type A receptors (GABA_A_Rs) in the mature brain, whereas in the early stages of development, GABA_A_R in this compartment is depolarizing/excitatory [1]. The depolarizing/excitatory action of GABA and GABA_A_R during brain development depends on the expression and activity of Na-K-2Cl cotransporter 1 (NKCC1), which is responsible for chloride influx in neurons. The shift to hyperpolarizing GABA_A_R-mediated responses during the second postnatal week is mainly due to the expression of the chloride exporter K-Cl cotransporter 2 (KCC2) [2]. However, the fact that NKCC1 is expressed throughout life in different cell types and subdomains [3,4] raises the question of its role in the mature brain.

At mature stages, NKCC1 regulates water flux and thereby cell volume, as well as extracellular electrolyte concentrations [5]. In addition, NKCC1 controls presynaptic neurotransmitter release in several neuronal types from different brain regions. For example, the presence of NKCC1 at the axon initial segment (AIS) of mature cortical principal neurons allows for the maintenance of depolarized reversal potential for GABA_A_R-mediated currents (E(GABA)) at axo-axonal synapses formed with GABAergic chandelier interneurons [6]. Other studies however show the appearance of a hyperpolarizing GABAergic polarity in the AIS during periadolescence [7] or in adulthood [8]. It has been proposed that GABA release from chandelier interneurons facilitates glutamate release from the presynaptic terminals of rat CA3 hippocampal pyramidal cells [9]. Similarly, NKCC1 is involved in the depolarization of presynaptic glutamatergic terminals in the hypothalamus [10], in the cerebellum [11] and at retinal photoreceptor synapses [12]. GABA-mediated depolarization enabled by NKCC1 Cl^−^ influx has also been shown to control corticotropin-releasing hormone secretion from the axon terminals in the medial eminence [13,14]. The peripheral axons of dorsal root ganglion neurons are also depolarized by GABA, which regulates the electrical excitability of unmyelinated C-fiber axons [15]. In the mature brain, NKCC1 is also expressed by non-neuronal cells, such as astrocytes [16,17]. In glioblastoma, the interaction of NKCC1 with cofilin via the regulation of actin dynamics may provide a mechanism for cell migration [18]. In the context of long-term potentiation, the interaction of NKCC1 with cofilin may regulate the dynamics of astrocytic processes via combined actions on the actin network and water influx, which may allow glutamate spillover and long-term potentiation [16]. Collectively, this suggests that NKCC1 may contribute in the mature brain to regulate water and ion flow, as well as neurotransmitter release and synaptic plasticity.

Key cellular and molecular mechanisms regulate the membrane stability of KCC2 in neurons [19,20]. Our team has shown the contribution of lateral diffusion in the rapid control of KCC2 membrane stability and chloride (Cl^−^) homeostasis in response to changes in neuronal glutamatergic and GABAergic activities [19,20,21,22]. KCC2 is diffuse in the plasma membrane, or it forms clusters around GABAergic and glutamatergic synapses [21]. Within the clusters, the transporters are anchored to the sub-membrane cytoskeleton by interacting with scaffolding molecules, such as the 4.1N protein [21]. Then, KCC2 transporters can escape from these clusters via lateral diffusion and reach clathrin-coated pits, where they are internalized and then recycled to the plasma membrane or degraded [21]. Transporters alternate between periods of slowing down and confinement within the clusters and periods of free movement outside the clusters. We have proposed that the “confined” and the “free moving” pools of transporters are in dynamic equilibrium, allowing the fine-tuning of the neuronal chloride level in membrane subdomains in response to local fluctuations in neuronal and synaptic activities [19,20]. Since activity-dependent changes in KCC2 mobility occur within tens of seconds [22], lateral diffusion is probably the first mechanism modulating KCC2 stability and function in the membrane. One question is whether similar mechanisms regulate NKCC1 on the neuronal surface. Here, we studied the distribution and diffusion of NKCC1a and NKCC1b on the surface of mature hippocampal neurons in comparison to those of KCC2. We found different diffusion behaviors that may explain the weaker clustering of NKCC1a/1b than that of KCC2 under basal activity conditions and its upregulation in the membrane under pathological conditions.

## 2. Material and Methods

### 2.1. Experimental Model and Subject Details

For all experiments performed on primary cultures of hippocampal neurons, animal procedures were carried out according to the European Community Council directive of 24 November 1986 (86/609/EEC), the guidelines of the French Ministry of Agriculture and the Direction Départementale de la Protection des Populations de Paris (Institut du Fer à Moulin, Animalerie des Rongeurs, license C 72-05-22). All efforts were made to minimize animal suffering and to reduce the number of animals used. Timed pregnant Sprague Dawley rats were supplied by Janvier Lab, and embryos were used at embryonic day 18 or 19 as described below.

### 2.2. Dissociated Hippocampal Cultures

Primary cultures of hippocampal neurons were prepared as previously described [21], with some modifications to the protocol. Briefly, hippocampi were dissected from embryonic day 18 or 19 Sprague Dawley rats of either sex. Tissue was then trypsinized (0.25% *v*/*v*) and mechanically dissociated in 1× HBSS (Invitrogen, Cergy Pontoise, France) containing 10mM HEPES (Invitrogen). Neurons were plated at a density of 120 × 103 cells/mL onto 18 mm diameter glass coverslips (Assistent, Winigor, Germany) pre-coated with 50 µg/mL poly-D,Lornithine (Sigma-Aldrich, Lyon, France) in plating medium composed of Minimum Essential Medium (MEM, Sigma-Aldrich) supplemented with horse serum (10% *v*/*v*, Invitrogen), L-glutamine (2 mM) and Na^+^ pyruvate (1 mM) (Invitrogen). After attachment for 3–4 h, cells were incubated in culture medium that consisted of neurobasal medium supplemented with B27 (1x, L-glutamine (2 mM) and antibiotics (penicillin 200 units/mL, streptomycin, 200 µg/mL)) (Invitrogen) for up to 4 weeks at 37 °C in a 5% CO_2_ humidified incubator. Each week, one-fifth of the culture medium volume was renewed.

### 2.3. DNA Constructs

The pcDNA3.1 Flag YFP hNKCC1 HA-ECL2 (NT931) was a gift from Biff Forbush (Addgene plasmid # 49,063; http://n2t.net/addgene:49063, accessed on 1 September 2018; RRID:Addgene_49063; [23]. From this NKCC1a-HA-Flag-mVenus plasmid, we raised the NKCC1a-HA-Δflag-ΔmVenus construct via the truncation of the tags located on NKCC1 NTD and the NKCC1b-HA-Δflag-ΔmVenus construct via the truncation of the splicing exon (CGAGGAAGAAGATGGCAAGACTGCAACTCAACCACTGTTGAAAAAAG). The following constructs were also used: pCAG_rat KCC2-3Flag-ECL2 [21], eGFP (Clontech, Saint-Germain-en-Laye, France), pCAG_GPHN.FingR-eGFP-CCR5TC [24] (gift from Don Arnold, Addgene plasmid # 46,296; http://n2t.net/addgene:46296; accessed on 1 September 2018, RRID:Addgene_46296), EYFP-Clathrin [25] (a gift from Xiaowei Zhuang, Addgene plasmid # 20921; http://n2t.net/addgene:20921 accessed on 1 September 2018; RRID:Addgene_20921), homer1c-DsRed and homer1c-GFP (kindly provided by D. Choquet, IIN, Bordeaux, France), and gephyrin-dendra2 (previously characterized and used in [26]). All constructs were sequenced by Beckman Coulter Genomics (Hope End, Takeley, UK).

### 2.4. Neuronal Transfection

Neuronal transfections were carried out at 13–14 DIV using transfectin (BioRad, Hercules, CA, USA), according to the manufacturers’ instructions (DNA: transfectin ratio 1 µg:3 µL), with 1–2 µg of plasmid DNA per 20 mm well. Simple transfections of NKCC1-HA-Flag-mVenus or KCC2-Flag were carried out with a plasmid concentration of 1 µg. The following ratios of plasmid DNA were used in co-transfection experiments: 0.4:0.4:0.4 µg for NKCC1a/b and KCC2 constructs together with GPHN.FingR-eGFP or homer1c-GFP for the quantification of clustering of NKCC1a/1b and NKCC1a/1b-KCC2 at synapses; 1:0.4:0.4 µg for NKCC1a/b constructs together with GPHN.FingR-eGFP and homer1c-DsRed for single-particle tracking (SPT) at synapses; 1:0.2 µg for NKCC1a/b or KCC2 constructs with eGFP for SPT on the AIS, axon and dendrite; 0.7:0.7 µg for NKCC1a/b or KCC2 constructs with clathrin-YFP or GPHN.FingR-eGFP for SPT in endocytic zones and STORM experiments, respectively; and 0.7:0.7 µg for NKCC1a/b or KCC2 constructs with dendra2-gephyrin for STORM/PALM. Experiments were performed 7–10 days post-transfection. SPT, STORM and STORM/PALM experiments were performed with Δflag-ΔmVenus NKCC1 constructs. Standard epifluorescence microscopy was performed with Flag-mVenus NKCC1 constructs.

### 2.5. Live Cell Staining for Quantum-Dot-Based Single-Particle Tracking

Neurons were stained as described previously [27]. Briefly, cells were incubated for 3–8 min at 37 °C with primary antibodies against HA tag (rabbit, 1:250, Cell signaling Technology, Saint-Cyr-L’École, France, Cat#3724) and Flag (mouse, 1:700, Sigma-Aldrich, Cat#F3165) for NKCC1 and KCC2 labeling, respectively. Antibodies were prepared in imaging medium, which consisted of phenol red-free minimal essential medium supplemented with glucose (33 mM; Sigma-Aldrich) and HEPES (20 mM), glutamine (2 mM), Na^+^-pyruvate (1 mM) and B27 (1x) from Invitrogen. After several washes with the imaging medium, the cells for NKCC1 detection were incubated for 1 min with F(ab’)2-Goat anti-Rabbit IgG (H + L) Secondary Antibody QDot emitting at 655 nm (1 nM; Invitrogen, Cat#10592815) in PBS (1 M; Invitrogen) supplemented with 10% Casein (*v*/*v*) (Sigma-Aldrich). The cells for KCC2 detection were incubated for 3–5 min at 37 °C with biotinylated Fab secondary antibodies (goat anti-mouse: 1:700; Jackson Immuno research, Montlucon, France, Cat#115-067-003, West Grove, PA, USA) in imaging medium. After washes, cells were incubated for 1 min with Qdot™ 655 Streptavidin Conjugate (1 nM; Thermo Fisher Scientific, Les Ulis, France, Cat#Q10123MP) in borate buffer (50 mM) supplemented with sucrose (200 mM) or in PBS (1 M; Invitrogen) supplemented with 10% Casein (*v*/*v*) (Sigma).

### 2.6. Staining on Fixed Cells

All KCC2 and NKCC1a/b staining procedures performed in this study were carried out on non-permeabilized cells. To preserve the neuronal membrane, the cultures were fixed for a very short time (4 min) with paraformaldehyde (4% *w*/*v*) supplemented with sucrose (20% *w*/*v*) to prevent cell damage. We tried different fixation times and ensured that the labeling of the cytoplasmic MAP2A protein was only visible in neurons permeabilized with 0.25% Triton.

### 2.7. Staining for STORM and STORM/PALM

For STORM of NKCC1 or KCC2 on the neuronal surface, cells were transfected at DIV14 with NKCC1a-HA-Δflag-ΔmVenus, NKCC1b-HA-Δflag-ΔmVenus or KCC2-3Flag-ECL2 constructs. They were then fixed at DIV21 for 4 min at room temperature (RT) in paraformaldehyde (PFA; 4% *w*/*v*; Sigma) and sucrose (20% *w*/*v*; Sigma) in 1× PBS. The cells were then washed in PBS and incubated for 30 min at RT in goat serum (GS; 3% *v*/*v*; Invitrogen) in PBS to block nonspecific staining. Neurons were then incubated for 120 min at RT with HA antibody (rabbit, 1:250, Cell signaling Technology, cat #C29F4) or Flag antibody (mouse, 1:400; Sigma, cat #F3165) in PBS-GS blocking solution. After washing, neurons were incubated with Alexa Fluor^®^ 647 AffiniPure Donkey Anti-Rabbit IgG (H + L) (2 µg/mL, Jackson ImmunoResearch, cat #711-605-152) or Alexa Fluor^®^ 647 AffiniPure Donkey Anti-Mouse IgG (H + L) (2 µg/mL, Jackson ImmunoResearch, cat #711-605-151) in PBS-GS solution.

For the STORM/PALM imaging of NKCC1 in relation with inhibitory synapses, cells were transfected at DIV14 with NKCC1a-HA-Δflag-ΔmVenus and dendra2-gephyrin construct. They were then fixed at DIV21 in 4% PFA for 15 min, washed in PBS 1x and stained for KCC2 or NKCC1 (as above).

### 2.8. Staining for Conventional Microscopy

For the surface staining of NKCC1 or KCC2, the staining was performed after a 4 min fixation at room temperature (RT) in paraformaldehyde (PFA; 4% *w*/*v*; Sigma) and sucrose (20% *w*/*v*; Sigma) solution in PBS 1x. The cells were then washed in PBS and incubated for 30 min at RT in goat serum (GS; 3% *v*/*v*; Invitrogen) in PBS to block nonspecific staining. Neurons were then incubated for 60–180 min at RT with HA antibody (rabbit, 1:250, Cell signaling Technology, Cat#C29F4) or Flag antibody (mouse, 1:400; Sigma, Cat#F3165) in PBS–GS blocking solution. After washing, neurons were incubated for 45–60 min with Cy3-Donkey Anti-Rabbit IgG (H + L) (min X) Secondary Antibody (1.9 µg/mL; Jackson ImmunoResearch, Cat#711-165-152) or Cy™3 AffiniPure Goat Anti-Mouse IgG (H + L) (1.9 µg/mL; Jackson ImmunoResearch, Cat#115-165-003) in PBS-GS, washed and finally mounted on glass slides using Mowiol 4-88 (48 mg/mL, Sigma).

For an analysis of membrane-associated NKCC1 in axons, cells were fixed for 4 min in PFA 4% + sucrose 20% and labeled for 60 min for HA (rabbit antibody, 1:250, Cell signaling Technology, cat #C29F4). HA antibody was then revealed with Cy3-Donkey Anti-Rabbit IgG (H + L) (min X) Secondary Antibody (1.9 µg/mL; Jackson ImmunoResearch, Cat#711-165-152). Cells were washed in PBS and permeabilized for 4 min at RT with Triton X-100 0.25% (*w*/*v*; Invitrogen) in PBS. Subsequently, neurons were incubated for 1 h with ankyrin antibody (mouse, 1:500, Merck Millipore, Guyancourt, France, Cat#MABN466). Ankyrin antibodies were revealed with Cy5-Donkey Anti-Mouse IgG (H + L) (min X) Secondary Antibody (1.9 µg/mL; Jackson Immuno Research, Cat# 715-175-150).

For the study of NKCC1 and KCC2 colocalization on the neuronal surface, cells were fixed for 4 min in PFA 4% + sucrose 20%, and then nonspecific sites were saturated for 30 min at RT in PBS-GS blocking solution. Cells were then incubated for 120 min at RT with HA antibody (rabbit, 1:500, Cell signaling Technology, Cat#C29F4) and Flag antibody (mouse, 1:400; Sigma, Cat#F3165) in PBS–GS blocking solution. Cells were then washed and incubated for 45 min with Cy™3 AffiniPure Goat Anti-Rabbit IgG (H + L) (1.9 µg/mL; Jackson ImmunoResearch, Cat#111-165-003) and Alexa Fluor^®^ 647 AffiniPure Donkey Anti-Mouse IgG (H + L) (1.9 µg/mL; Jackson ImmunoResearch, Cat#715-605-150) antibodies in PBS-GS solution. After washes, cells were mounted on glass slides using Mowiol 4-88 (48 mg/mL, Sigma).

For the detection of GABA_A_Rs in clathrin-coated pits, the GABA_A_R γ2 subunits were stained by incubating living neurons for 20 min at 4 °C with primary antibodies against extracellular epitopes of GABA_A_R γ2 subunit (guinea pig: 2 µg/mL, Synaptic Systems, Cat#224004) diluted in imaging medium. After three washes in imaging medium, neurons were fixed for 15 min at room temperature (RT) in paraformaldehyde (PFA, 4% *w*/*v*, Sigma) and sucrose (20% *w*/*v*, Sigma) solution prepared in PBS (1x). Cells were washed in PBS and incubated for 30 min at RT in normal goat serum (GS, 3% *v*/*v*, Invitrogen) in PBS to block nonspecific staining. Cells were then incubated for 60 min at RT with CY3-conjugated donkey anti-guinea pig (3.75 µg/mL, Jackson Immunoresearch, Cat#706-165-148) in PBS-GS blocking solution, washed and finally mounted on glass slides using Mowiol 4-88 (48 mg/mL, Sigma).

In all experiments, the sets of neurons compared for quantification were labeled and then imaged simultaneously.

## 3. Quantification and Statistical Analysis

### 3.1. Single-Particle Tracking and Analysis

Cells were imaged using an Olympus IX71 inverted microscope equipped with a 60X objective (NA 1.42; Olympus, Rungis, France) and a 120 W Mercury lamp (X-Cite 120Q, Lumen Dynamics, Zaandam, The Netherlands). Individual images of gephyrin-YFP and homer1c-DsRed, and QD real-time recordings (integration time of 30 ms over 1200 consecutive frames) were acquired with an ImagEM EMCCD (Hamamatsu, Massy, France) camera and MetaView software (Meta Imaging 7.7). The cells were imaged within 45 min following appropriate drug pre-incubation. QD tracking and trajectory reconstruction were performed with homemade software (Matlab R2017a; The Mathworks, Natick, MA, USA) as described in [27]. One to two sub-regions of dendrites were quantified per cell. In cases of QD crossing, the trajectories were discarded from the analysis. The reconstructed trajectories were considered synaptic when they colocalized with a binary mask formed by fluorescent clusters of GPHN.FingR-eGFP and homer1c-DsRed, two postsynaptic markers enriched at inhibitory and excitatory synapses, respectively, or they were considered extrasynaptic when they were located two pixels (380 nm) apart. The values of the mean square displacement (MSD) plot vs. time were calculated for each trajectory by applying the following relation:$$MSD(n\tau )=\frac{1}{N-n}\sum_{i=1}^{N-n}\left[(x_{i+n}-x_{i})²+(y_{i+n}-y_{i})²\right]$$
where τ is the acquisition time; N is the total number of frames; and n and i are positive integers, with n determining the time increment. Diffusion coefficients (D) were calculated by fitting the first four points without the origin of the MSD vs. time curves with the equation MSD(nτ) = 4Dnτ + σ, where σ is the spot localization accuracy. Depending on the type of lamp used for imaging, the QD pointing accuracy was ~20–30 nm, a value well below the measured explored areas (at least 1 log difference). The explored area of each trajectory was defined as the MSD value of the trajectory at two different time intervals of 0.42 and 0.45 s [28]. The synaptic dwell time was defined as the duration of detection of QDs at/near synapses on a recording divided by the number of exits, as detailed previously [27]. Dwell times < 5 frames were not retained.

### 3.2. Fluorescence Image Acquisition and Analysis

The sets of neurons compared for quantification were imaged simultaneously. Image acquisition was performed using a ×100 objective (NA 1.40) on a Leica (Nussloch, Germany) DM6000 upright epifluorescence microscope with a 12-bit cooled CCD camera (Micromax, Roper Scientific, Evry, France) run using MetaMorph software Series 7.8 (Roper Scientific). Quantification was performed using MetaMorph software (Roper Scientific) (Appendix A). To assess NKCC1 and KCC2 clusters, the exposure time was fixed at a non-saturating level and kept unchanged between cells and conditions. For each neuron, a well-focused dendrite was chosen, and an ROI surrounding the dendrite was manually selected. Only primary dendrites were considered for the cluster analysis. The region of interest was started after the cell body and the dendrite of ~20–130 µm length were considered. The images were then flattened, the background was filtered (kernel size, 3 × 3 × 2) to enhance cluster outlines, and a user-defined intensity threshold was applied to select clusters and avoid their coalescence. Only clusters ≥ 2 pixels were taken into account for the analysis. Clusters were then outlined, and the corresponding regions were transferred onto raw images to determine the mean cluster number, area and fluorescence intensity. Then, the dendritic length of the ROI was measured to determine the number of NKCC1 or KCC2 clusters per 10 µm. For the quantification of synaptic NKCC1 clusters or for the study of NKCC1 and KCC2 colocalization, NKCC1 clusters colocalized on at least 1 pixel with gephyrin-GFP, homer-GFP clusters or KCC2 clusters were considered. For each culture, we analyzed ~10–18 cells per experimental condition and ~20–100 clusters per cell. Due to the variability in synapse density between cultures, the number of clusters per 10 µm in each culture was normalized to the respective control values, allowing for comparisons between cultures.

### 3.3. STORM and PALM Microscopy

Samples were mounted in a Ludin chamber and imaged in a PBS-based oxygen-scavenging buffer containing an imaging buffer (Tris 100 mM, NaCl 20 mM, pH 8), glucose 40%, PBS 1x, MEA (HCl, cysteamine; Sigma), catalase 5 mg/mL (Sigma) and glucose oxidase 200 U/mL (Sigma). Catalase was diluted in MgCl 4 mM, 2 mM EGTA and PIPES 24 mM (Sigma, pH 6.8). Pyranose oxidase was diluted in the same buffer supplemented with glycerol (50% *v*/*v*). Imaging was performed in a semi-total internal reflection fluorescence (TIRF) mode on an inverted N-STORM Nikon Eclipse Ti microscope with a 100× oil immersion objective (NA 1.49) and an Andor iXon Ultra EMCCD (Oxford Instruments, Les Ulis, France) camera (image pixel size, 160 nm), using specific lasers for the STORM imaging of Alexa647 (640 nm) and for the PALM imaging of dendra2 (405 and 561 nm). KCC2 or NKCC1a and NKCC1b videos of 30,000 frames were acquired at a frame rate of 50 ms. Gephyrin-dendra2 movies of ~20,000 frames were acquired at a frame rate of 20 ms. The z position was maintained during acquisition by a Nikon (Lisses, France) perfect focus system. For the comparison of KCC2 and NKCC1 clustering using STORM, the labeling and image acquisition for NKCC1a, NKCC1b and KCC2 were performed in parallel; i.e., all the samples were treated the same way. We also made sure that we obtained almost all single molecules by progressively increasing the 405 nm laser power from 1% up to 9% during the recording session.

Single-molecule localization and 2D image reconstruction were conducted as described in [29] by fitting the PSF of spatially separated fluorophores to a 2D Gaussian distribution. The positions of the fluorophores were corrected by the relative movement of the cluster by calculating the center of mass of the cluster throughout the acquisition using a partial reconstruction of 2000 frames with a sliding window [29]. STORM and PALM images were rendered by superimposing the coordinates of single-molecule detections, which were represented with 2D Gaussian curves of unitary intensity. To correct multiple detections of the same molecule, we identified detections occurring in the vicinity of space (2σ) and time (15 s) as belonging to the same molecule.

The surfaces of NKCC1 and KCC2 clusters and the densities of molecules per square nanometer were measured in reconstructed 2D images through cluster segmentation based on detection densities. The minimal thresholds used to determine clusters were 1% intensity, 0.1 per nm^2^ minimum detection density and 10 detections. The resulting binary image was analyzed with the function “regionprops” of Matlab to extract the surface area of each cluster identified using this function. The density was calculated as the total number of detections in the pixels (pixel size = 20 nm) belonging to a given cluster, divided by the area of the cluster.

### 3.4. Statistics

Sampling corresponds to the number of quantum dots for SPT and the number of cells for ICC. Sample size selection for experiments was based on published experiments, pilot studies and in-house expertise. All results were used for analysis, except for a few cases. For the imaging experiments (SPT, immunofluorescence), cells with reduced cell viability (apparition of blobs, fragmented neurites) were discarded from the analysis. In most cases, data representation was performed with boxplots or cumulative frequency plots. The statistical test used to compare two groups was the Welch t-test when the normality assumption was met (Q-Q plots and cumulative frequency fit); otherwise, the Mann–Whitney test was performed to assess whether there was a presence of dominance between the two distributions. For variables following a log-normal distribution, such as the variables obtained from the SPT and STORM assays, we applied the log(.) function after division by the control group’s median value. For super-resolution experiments, as an important variability could be observed between different cells in the same coverslip, a balanced random selection of clusters across neurons, conditions and cultures was performed, and then the variables from each culture were divided by the median of the control group. The results from different cultures were pooled, and log(.) was applied; then, the Mann–Whitney U value was computed. This process was repeated 1000 times, and the *p*-value was determined from the U distribution using the basic definition of the *p*-value. For SPT analysis, note that each QD was associated with 3 EA values; thus, the sample size was 3 times greater. Statistical analyses were performed with R version 3.6.1 (Package used: ggplot2, matrixStats, R core Team, Vienna, Austria). Statistical tests were performed between a condition and its control only using cultures where both conditions were tested. For the calcium imaging analysis, statistics (paired *t*-test) were run for each cell on the mean fluorescence intensities calculated before and after drug application (all time points included). Differences were considered significant for *p*-values less than 5% (* *p* ≤ 0.05; ** *p* < 0.01; *** *p* < 0.001).

## 4. Results

### 4.1. NKCC1a and NKCC1b Are Targeted to Somato-Dendritic and Axonal Compartments of Mature Hippocampal Neurons

Two splicing variants of NKCC1 have been described: NKCC1a and NKCC1b [30]. NKCC1b differs from NKCC1a by the absence of an exon that contains a putative consensus sequence for PKA phosphorylation [30] and a dileucine motif essential for the basolateral sorting of NKCC1a in polarized epithelial cells [31]. In the adult human brain, NKCC1b transcripts are detected at a higher level than NKCC1a [32]. In the mouse cortex and hippocampus, NKCC1b transcripts predominate in neurons, while NKCC1a transcripts are found in non-neuronal cells [33]. However, there is currently no antibody specific to either isoform that can correlate protein expression with transcript expression. In the absence of such antibodies, we transfected primary cultures of hippocampal neurons at 14 days in vitro (DIV) with NKCC1a or NKCC1b constructs bearing a 2xHA tag at position Histidine^398^ in the second extracellular loop to study the surface expression, distribution and diffusion of the two NKCC1 isoforms (see Section 2). Neurons were co-transfected with KCC2-Flag constructs for comparison. The membrane staining of HA and Flag was performed at 21 DIV. These 21-DIV-old hippocampal neurons differentiate structurally and functionally mature GABAergic inhibitory and glutamatergic synapses, as shown by the formation of GABA_A_R nanodomains at postsynaptic sites [34], the differentiation of glutamatergic synapses on dendritic spines [35] and the recording of AMPAR-mediated mEPScs [35] and of hyperpolarizing [22] GABA_A_R-mediated mIPSCs [36]. We also reported that Flag-tagged constructs of KCC2 are functional and are targeted to the somato-dendritic membrane as the endogenous protein [21]. Moreover, we recently showed that transfecting hippocampal neurons with HA-tagged NKCC1 has no main impact on the intracellular chloride concentration [Cl^−^]_i_ under basal activity conditions [37], suggesting that transfected neurons do not massively overexpress Flag-tagged KCC2 or HA-tagged NKCC1.

First, we measured the surface expression levels of exogenous NKCC1a and NKCC1b and compared them to those of KCC2 by analyzing the average intensity per pixel of HA and Flag tags in a dendritic region of interest (ROI). We found that KCC2 labeling on the neuronal surface was 1.37 ± 0.1-fold (Mann–Whitney rank sum test *p* = 2.0 × 10^−3^) and 1.46 ± 0.1-fold (*p* < 1.0 × 10^−3^) higher than that of NKCC1a and NKCC1b, respectively (from 50–57 cells, three independent cultures). As expected, mature 21-DIV-old hippocampal neurons expressed significantly more recombinant KCC2 than NKCC1a and NKCC1b on their surface, indicating that neurons adapt the membrane level of Flag-tagged KCC2 and HA-tagged NKCC1.

We then reported that, similarly to KCC2 (Figure 1(A1–A3)), NKCC1a and NKCC1b are targeted to the somato-dendritic plasma membrane (Figure 1(B1–B3,C1–C3), respectively), where they form clusters (e.g., white arrowheads in Figure 1(B2,B3,C2,C3), respectively). This somato-dendritic clustering of NKCC1a and NKCC1b is similar to that of KCC2 (Figure 1(A2,A3)). The fact that NKCC1a and NKCC1b were both detected in the neuronal cell body and along dendrites suggests that the dileucine motif is not essential for the somato-dendritic localization of the transporter.

We next assessed the localization of NKCC1a, NKCC1b and KCC2 in the axon initial segment (AIS) and along the axon. We transfected neurons with NKCC1a-HA, NKCC1b-HA or KCC2-Flag constructs together with cytoplasmic eGFP. Cytoplasmic eGFP filled the whole neuron, including the soma, dendrites, the AIS and axons (Figure 1(A1,A3,B1,B3,C1,C3)). Axons were distinguished from dendrites by the fact that they emerged from the cell body as thinner and longer neurites (Figure 1(A3,B3,C3)). We found that KCC2 clustering was high in the cell body and in dendrites (white arrowheads in Figure 1(A2,A3)) but absent from the eGFP-positive AIS and axon (yellow arrowheads in Figure 1(A2,A3)). In contrast, NKCC1a and NKCC1b clusters could be detected in the somato-dendritic compartment but also along the thin eGFP-labeled axon (Figure 1(B2,B3, C2,C3), respectively). To confirm the axonal targeting of NKCC1 but not that of KCC2, we performed double-staining experiments of HA or Flag tags with ankyrin, a marker of the AIS (Figure 1(D1–E2)). Ankyrin accumulated in the first ~10–100 μm of the axon (Figure 1(D2,E2)). For the analysis of KCC2 in the axon, cells were sequentially labeled for KCC2-Flag (surface staining), permeabilized and then stained for ankyrin. This was necessary because both Flag and ankyrin antibodies were produced in mice, resulting in secondary antibody (Cy5-coupled donkey anti-mouse and Cy3-coupled goat anti-mouse) cross-reactivity. The resulting immunoreactivity showed a Flag-positive neuron (Figure 1(D1)) with nonspecific Cy5 staining (white in the soma and dendrites of the neuron in Figure 1(D2)), which is due to Flag labeling with Cy3- and Cy5-coupled secondary antibodies. Such cross-reactivity did not occur between rabbit anti-HA and mouse anti-ankyrin in the double labeling of NKCC1 and ankyrin (Figure 1(E1,E2)). Despite this cross-reactivity in the soma and dendrites, which highlights the presence of KCC2 in these compartments, it should be noted that no cross-reactivity was observed in the AIS (blue arrowheads in Figure 1(D2)), thus showing that KCC2 was absent from the AIS. In contrast, NKCC1a formed clusters in the somato-dendritic compartment (white arrowheads in Figure 1(E1,E2)) and in the AIS (blue arrowheads in Figure 1(E1,E2)). The fact that NKCC1a but not KCC2 could be found in the AIS and along the axon suggests a specific targeting and probably a role for NKCC1 in the axon.

### 4.2. NKCC1a and NKCC1b Are Co-Clustered with KCC2 on the Dendrite

Next, we investigated the distribution of dendritic NKCC1a or NKCC1b clusters relative to KCC2 clusters and to GABAergic and glutamatergic synapses in neurons transfected with NKCC1a-HA or NKCC1b-HA together with KCC2-Flag and either gephyrin.FingR-eGFP or homer1c-GFP to label inhibitory and excitatory synapses, respectively (Figure 2). We observed that many NKCC1a or NKCC1b dendritic clusters were partially or completely superimposed on KCC2 clusters (Figure 2A,B). Interestingly, the NKCC1a or NKCC1b clusters that colocalized with KCC2 clusters (white arrows in Figure 2A,B) appeared to be larger than non-colocalized clusters (red arrows in Figure 2A,B). Some (white arrows) but not all (orange arrows) of these NKCC1a/b-KCC2 clusters were at or near GABAergic (Figure 2A) or glutamatergic (Figure 2B) synaptic markers. No main difference could be observed between NKCC1a and NKCC1b in terms of the overall proportion of clusters colocalized with KCC2 or at/near synapses (Figure 2A,B).

We quantified the clustering of NKCC1a and NKCC1b on the surface of dendrites and compared it to that of KCC2 (Figure 3A–C, respectively). When compared to each other, the density of NKCC1b clusters per unit length was slightly higher than that of NKCC1a (Figure 3D). However, NKCC1a and NKCC1b clusters were of similar size (Figure 3E), pixel intensity (Figure 3F) and cluster intensity (Figure 3G), indicating similar clustering properties for the two splicing isoforms on the dendritic surface. When compared to KCC2 clusters, NKCC1a and NKCC1b clusters were smaller (Figure 3E). The intensity per pixel of NKCC1a and NKCC1b clusters was lower than that of KCC2 clusters (Figure 3F), suggesting less compaction of the molecules within clusters. Furthermore, the total number of molecules per cluster was also reduced for NKCC1a and NKCC1b compared to KCC2, as revealed by the lower fluorescence intensity of these clusters (Figure 3G). These differences in fluorescence intensity between NKCC1a/b and KCC2 are not related to the choice of fluorophores used, since NKCC1a/b were revealed with secondary antibodies coupled to a brighter fluorophore than KCC2 (CY3 vs. Alexa Fluor647). Therefore, this reduced clustering of NKCC1a and NKCC1b relative to that of KCC2 is consistent with the idea that NKCC1 transporters are less abundant than KCC2 (see above) on the dendritic surface.

Based on our observations in Figure 2, we quantified the colocalization of NKCC1a or NKCC1b with KCC2 (see Section 2 and Appendix A). We found that 58.8 ± 2.4% of NKCC1a and 56.7 ± 2.6% of NKCC1b clusters were colocalized with KCC2 clusters (Figure 3H). Interestingly, the clusters of NKCC1a or NKCC1b that colocalized with those of KCC2 were ~1.6 and ~1.8 times bigger than the non-colocalized ones (Figure 3I). The mean pixel intensity per cluster was the same for the colocalized and non-colocalized clusters (Figure 3J), suggesting a similar compaction of molecules within clusters. However, the NKCC1a and NKCC1b clusters that colocalized with KCC2 were ~1.6 and ~2.0 times more intense than the non-colocalized ones (Figure 3K), highlighting an increased number of NKCC1a and NKCC1b molecules in clusters where KCC2 also accumulated. As a result, the clustering of NKCC1a or NKCC1b was higher in NKCC1a/b-KCC2 clusters than it was in the absence of KCC2. However, to be sure that NKCC1a/b-KCC2 colocalization was not random, we calculated the surface area occupied by the dendritic clusters of KCC2 as well as the surface area occupied by the dendritic clusters of NKCC1a or NKCC1b colocalized or not with KCC2 (Appendix A). We found that KCC2 clusters occupied 4.1 ± 0.5% of the surface of the dendritic ROI, and NKCC1a clusters that colocalized or not with KCC2 occupied 2.5 ± 0.5% and 1.0 ± 0.1%, respectively, of this dendritic surface (37 cells, two cultures). Similar results were obtained for NKCC1b: KCC2 clusters occupied 3.7 ± 0.3% of the dendritic surface, and NKCC1b clusters that colocalized or did not colocalize with KCC2 occupied 2.3 ± 0.2% and 1.2 ± 0.2%, respectively, of the surface of the dendritic ROI studied (39 cells, two cultures). Thus, NKCC1a and NKCC1b transporters occupy a small surface area of the dendrite regardless of whether they are colocalized with KCC2. As our results demonstrate specific features of the NKCC1a/b-KCC2 clusters colocalized with KCC2 (Figure 3H–K), this enables us to propose that the colocalization of NKCC1a/b-KCC2 clusters is not coincidental.

### 4.3. Synaptic Targeting of NKCC1a and NKCC1b

Next, we quantified the proportion of NKCC1a or NKCC1b clusters at inhibitory GABAergic and excitatory glutamatergic synapses and determined their characteristics relative to those of non-synaptic clusters from images of NKCC1a or NKCC1b and gephyrin.FingR-eGFP or homer1c-GFP (as in Figure 2). The synaptic distributions of NKCC1a and NKCC1b clusters were comparable. A subpopulation (~30%) of NKCC1a or NKCC1b clusters was at/near synapses (Figure 3L). The proportion of NKCC1a and NKCC1b clusters at/near excitatory synapses was higher than that at/near inhibitory synapses (17.8 ± 1.2% vs. 11.3 ± 1.0% and 15.8 ± 1.3% vs. 12.7 ± 1.2% for NKCC1a and NKCC1b respectively, Figure 3L). Next, we compared synaptic NKCC1a or NKCC1b clusters to their respective extrasynaptic clusters (Figure 3M–O). We found that NKCC1a or NKCC1b clusters at/near glutamatergic or GABAergic synapses were significantly larger (Figure 3M) and more intense (Figure 3O) than those detected far from synapses, although their compaction did not change as indicated by their similar pixel intensity (Figure 3N). We concluded that NKCC1a and NKCC1b are more efficiently recruited within clusters at/near synapses than in non-synaptic regions.

### 4.4. Special Features of NKCC1a, NKCC1b and KCC2 Clusters Revealed Using STORM

Since NKCC1a, NKCC1b and KCC2 clusters are small (e.g., Figure 3A–C, respectively), we further compared NKCC1a/b and KCC2 clustering using super-resolution STORM. NKCC1a (Figure 4A–C), NKCC1b (Figure 4D–F) and KCC2 (Figure 4G–I) clusters were evenly distributed along the dendritic shaft (Figure 4B,E,H, respectively) and in dendritic spines (Figure 4C,F,I, respectively). Most NKCC1a and NKCC1b clusters were round (arrowheads in Figure 4A,D), while KCC2 clusters displayed round (arrowheads in Figure 4G) or elongated shapes (arrows in Figure 4G). We quantified these observations. The density of NKCC1b clusters along the dendrites was lower than that of KCC2 (Figure 4J). A similar trend was observed for NKCC1a, although this difference was not significant (Figure 4J). STORM reported that NKCC1a and NKCC1b clusters were smaller than KCC2 clusters (Figure 4K, although not significant for NKCC1a). The mean surface areas of NKCC1a and NKCC1b clusters were 90.75 ± 3.67 nm^2^ and 86.96 ± 3.28 nm^2^ (mean ± SEM), while KCC2 clusters had an average surface area of 143.06 ± 6.38 nm^2^
*(*Figure 4K). The number of single molecules detected per cluster (Figure 4L) was smaller for NKCC1a and NKCC1b clusters than for KCC2 clusters, indicating fewer NKCC1a/b molecules per cluster. The density of molecules per cluster was also lower for NKCC1a/b clusters than for KCC2 clusters (Figure 4M), suggesting reduced molecular compaction within the cluster for NKCC1a/b. These data are in agreement with our results obtained with conventional microscopy showing a smaller size, a lower pixel intensity per cluster and a lower cluster intensity for NKCC1a and NKCC1b than for KCC2 (Figure 3I–K, respectively). Therefore, NKCC1a and NKCC1b clustering is lower than KCC2 clustering. Interestingly, STORM also revealed a difference in NKCC1a and NKCC1b dendritic clustering: NKCC1a clusters were larger (Figure 4K), contained more molecules (Figure 4L) and were more compact than NKCC1b clusters (Figure 4M), suggesting higher clustering for NKCC1a.

### 4.5. NKCC1a and KCC2 Clusters Are Not Postsynaptic but Surround Inhibitory Synapses

To study the precise relationship of NKCC1a with inhibitory synapses, we performed STORM of NKCC1a-HA labeling and PALM analysis of dendra2-gephyrin. Dendra2-gephyrin detected inhibitory subsynaptic domains in the dendritic shafts and in some spines of mature hippocampal neurons (Figure 5A–E). This is reminiscent of the nanoscopic organization of inhibitory postsynaptic densities shown using super-resolution [26,38]. NKCC1a clusters were present in dendritic shafts and spines (Figure 5A). Many of these clusters were far from dendra2-gephyrin subsynaptic domains (arrowheads in Figure 5A), indicating that they are in the extrasynaptic membrane and/or at/near excitatory glutamatergic synapses formed on dendritic spines (arrow in Figure 5A). However, NKCC1a clusters could also be detected near dendra2-gephyrin subsynaptic domains (Figure 5A–C). Interestingly, NKCC1a clusters did not colocalize with gephyrin subsynaptic domains but were distributed around these domains (Figure 5B,C). This indicates a perisynaptic rather than postsynaptic localization of NKCC1a. This distribution of NKCC1a is similar to that of KCC2: KCC2-Flag STORM and dendra2-gephyrin PALM revealed the presence of KCC2 clusters in dendritic shafts (arrowheads in Figure 5D) and spines (arrows in Figure 5D) distal to inhibitory synapses, as well as surrounding inhibitory synapses (Figure 5D,E). Therefore, NKCC1a and KCC2 are not found within the postsynapse, but they accumulate at the periphery of inhibitory synapses.

### 4.6. NKCC1a and NKCC1b Are More Mobile Than KCC2 on the Surface of Dendrites

NKCC1a/b clustering suggests the involvement of the diffusion–capture mechanism in the regulation of these aggregates, as shown for KCC2 [21,22] and neurotransmitter receptors [39,40]. Such a mechanism may allow for the rapid control of the number of transporters in the membrane subdomains of neurons and, thus, the rapid control of chloride levels.

We first compared the membrane dynamics of NKCC1a and NKCC1b using the quantum-dot-based single-particle tracking (QD-SPT) technique in mature hippocampal neurons. To this end, neurons were transfected at DIV 14 with recombinant HA-tagged NKCC1a or NKCC1b and stained live at DIV 21–24 with a QD-coupled anti-HA antibody, and real-time imaging was performed (see Section 2). Then, NKCC1a and NKCC1b trajectories were reconstructed with homemade software, and the diffusion coefficient and the size of the explored area were extracted for each trajectory from the mean square displacement (MSD) plot versus time function. The observation of individual trajectories showed that NKCC1a and NKCC1b explored a large surface area of the somato-dendritic membrane (Figure 6A). A quantitative analysis performed on the whole (synaptic + extrasynaptic) population of trajectories revealed that the median value of the diffusion coefficient (Figure 6B) and the median value of the explored area (Figure 6C) were not significantly different between NKCC1a and NKCC1b, indicating similar diffusion properties. In comparison with KCC2 trajectories, we found that NKCC1a and NKCC1b trajectories were longer and extended over a larger area of the membrane (Figure 6A), suggesting reduced diffusion constraints for NKCC1a and NKCC1b in the plasma membrane. This was confirmed by the quantitative analysis showing a greater diffusion coefficient (Figure 6D) and explored area (Figure 6E) for NKCC1a than for KCC2 on the dendritic surface. Our data indicate that NKCC1a and NKCC1b are less restricted in their movement in the dendritic membrane than KCC2, likely reflecting a weaker cytoskeletal anchoring of NKCC1 transporters than that of KCC2.

### 4.7. NKCC1a Is Confined at/near Synapses

We studied the diffusive behavior of NKCC1a on the dendrites of neurons co-transfected with NKCC1a-HA and the inhibitory and excitatory synaptic markers gephyrin-FingR-eGFP (Figure 7A–C,G) and Homer1c-DsRed, respectively (Figure 7D–G). We found that individual QDs explored a restricted area of the membrane near GABAergic (Figure 7A–C,G) and glutamatergic (Figure 7D–G) synapses compared with extrasynaptic regions (Figure 7G). However, we observed that NKCC1a sometimes escaped the perisynaptic region to explore neighboring synapses regardless of whether they were excitatory or inhibitory synapses (Figure 7G). This is reminiscent of the diffusive behavior of KCC2 [21,22].

We next measured the synaptic and extrasynaptic mobility of NKCC1a on hundreds of trajectories. We reported a reduced slope of the MSD vs. time function for perisynaptic trajectories as compared with extrasynaptic ones (Figure 7H), indicative of an increased confinement of NKCC1a at the periphery of both excitatory and inhibitory synapses. Consistently, the median values of the diffusion coefficient and explored area were also significantly decreased at the periphery of glutamatergic and GABAergic synapses (Figure 7I,J). Therefore, NKCC1a is confined at the periphery of synapses. Although the diffusion coefficient and the size of the explored area of NKCC1a were similar near both excitatory and inhibitory synapses, the synaptic dwell time of NKCC1a near glutamatergic synapses was ~1.9-fold higher than that of the transporter near inhibitory synapses (Figure 7K). Thus, NKCC1a is more confined near glutamatergic synapses than near GABAergic synapses. This is reminiscent of KCC2 mobility at/near synapses [21].

### 4.8. NKCC1a/1b Are Confined within Endocytic Zones

As NKCC1a and NKCC1b are expressed on the dendritic membrane at a lower level than KCC2 (see above), we wondered whether NKCC1a and NKCC1b had a particular relationship with endocytic vesicles, where they would be internalized, or whether they could be transiently stored before being released into the membrane pool. These “endocytic zones” would therefore constitute a reserve pool of transporters that could be rapidly mobilized to the membrane according to demand. We analyzed the diffusion properties of dendritic NKCC1a and NKCC1b in endocytic zones identified by the presence of clathrin-YFP clusters in transfected neurons (Figure 8A). Clathrin-YFP form numerous aggregates all along the dendrites, with some of them being adjacent to GABA_A_R clusters (Figure 8A), in agreement with the notion that GABAergic synapses are surrounded by clathrin-coated pits [41]. Individual NKCC1a trajectories exhibited decreased surface exploration in clathrin-YFP clusters compared to remotely located regions (Figure 8B). This was consistent with the significant decrease in the diffusion coefficients (Figure 8C) and explored area (Figure 8D) of NKCC1a in endocytic zones as compared to transporters detected far from clathrin-YFP clusters. NKCC1b showed the same tendency as NKCC1a to slow down and confine to the endocytic zones, although the differences between inside and outside the endocytic zones were not significant (Figure 8C,D). A comparison of NKCC1a and NKCC1b revealed that NKCC1a was slower (Figure 8C) and more confined to endocytic zones than NKCC1b (Figure 8D), while they displayed similar values of the diffusion coefficient and surface exploration outside endocytic zones (Figure 8C,D). This suggests that NKCC1a is more efficiently trapped within endocytic zones than NKCC1b. As noticed for the overall mobility (Figure 6D,E), the diffusion coefficient (Figure 8C) and surface area explored (Figure 8D) by KCC2 were significantly lower in clathrin-coated pits, as well as outside the endocytic zones, compared to NKCC1a and NKCC1b. Thus, NKCC1a and NKCC1b are more mobile in the plasma membrane than KCC2 regardless of the membrane compartment. Unlike NKCC1a and NKCC1b, the diffusion coefficient (Figure 8C) and the size of the explored area (Figure 8D) of KCC2 did not differ inside and outside clathrin-YFP clusters, indicating that, at least under basal activity conditions, KCC2 confinement is the same inside and outside endocytic zones. Interestingly, although KCC2 was significantly less mobile than NKCC1 on the dendrite, NKCC1a spent significantly more time in endocytic zones than KCC2 (Figure 8E). Taken together, our results indicate that NKCC1a and NKCC1b transporters are more efficiently retained within endocytic zones than KCC2. When comparing NKCC1a dwell time in the endocytic zone vs. at/near inhibitory synapses, we found that NKCC1a spent more time in inhibitory synapses than in endocytic zones (Figure 8E).

## 5. Discussion

Here, we studied in hippocampal neurons the cellular and subcellular distributions, as well as the lateral diffusion, of the cotransporters NKCC1a and NKCC1b and compared them to those of KCC2. In the absence of antibodies targeting NKCC1a and NKCC1b on the surface of neurons, we transfected neurons with HA-tagged constructs (see Section 2). Recombinant protein expressions have been used to study the cell biology of KCC2 and regulation by the neuronal activity of this transporter [21,22]. The cellular and subcellular distributions of exogenous and endogenous KCC2 are comparable [21]. We then reported fine regulation via the glutamatergic and GABAergic transmission of KCC2 diffusion and clustering via calcium and chloride signaling pathways specific for glutamatergic and GABAergic transmission [21,22]. Using a similar strategy, we show here, as expected for NKCC1 in 21-DIV-old mature neurons, that HA-tagged NKCC1a and NKCC1b are less abundant than Flag-tagged KCC2 on the surface of neurons. Thus, NKCC1a and NKCC1b are not highly overexpressed in neurons. Moreover, they have a different distribution from KCC2-Flag: they are detected in the AIS, axon, soma and dendrites, while KCC2 is excluded from the axonal compartment. We found that NKCC1a and NKCC1b transporters are organized in clusters in the axonal and somato-dendritic membranes. We compared the membrane expression and clustering of NKCC1a and NKCC1b using standard microscopy and STORM. Our data (Figure 3 and Figure 4) show little difference between NKCC1a and NKCC1b. Both isoforms were detected at lower levels than KCC2 on the surface of neurons and formed fewer, smaller and less packed clusters than KCC2. A similar proportion of these clusters was colocalized with KCC2 (~55%) and was found at synapses (~30% at glutamatergic and GABAergic synapses). However, STORM showed a difference between NKCC1a and NKCC1b: NKCC1a clusters were slightly larger and more densely packed than NKCC1b clusters. A comparison of NKCC1a and NKCC1b lateral diffusion revealed that NKCC1a was slower and more confined to endocytic zones than NKCC1b, while they displayed similar values of the diffusion coefficient and surface exploration outside endocytic zones, suggesting that NKCC1a is more efficiently trapped within endocytic zones than NKCC1b. It is tempting to reconcile the SPT data with the STORM data and propose that the higher confinement of NKCC1a relative to that of NKCC1b in endocytic zones reflects a greater proportion of NKCC1a stored in the endocytic zones than NKCC1b. The pool of endocytic transporters could be internalized, or it could, under certain circumstances, escape the endocytic zones to increase the amount of transporter in the plasma membrane.

In the dendrite, some NKCC1a and NKCC1b transporters accumulated at the periphery of excitatory glutamatergic and inhibitory GABAergic synapses, where they colocalized with KCC2 clusters. Interestingly, these NKCC1a/b-KCC2 clusters were bigger and denser in molecules than the non-colocalized ones. To the best of our knowledge, this is the first time that the colocalization of NKCC1a/b and KCC2 was observed in neurons. Homodimers of NKCC1-NKCC1 and KCC2-KCC2 have been characterized, as well as the dimerization domains identified in the respective C-terminal regions of NKCC1 and KCC2 (reviewed in [42]). Although NKCC1-KCC4 heterodimerization has been reported in oocytes (Simard et al., 2007), NKCC1-KCC2 heteromerization is not known to date. This NKCC1a/b-KCC2 co-clustering suggests however a physical interaction between the two cotransporters. This remains to be elucidated. In any case, this co-distribution is not a coincidence, as it concerns a large majority (>50%) of NKCC1a/b clusters that occupy a small surface area (<5%) of the dendritic membrane, and these NKCC1a/b-KCC2 clusters are particularly well structured with a much larger size and density of molecules than NKCC1a/b clusters not colocalized with KCC2. What is the interest of such proximity between these transporters? Knowing that some of these co-clusters are present at the periphery of synapses, it could be assumed that NKCC1 participates as KCC2 in the following: (i) the local regulation of chloride homeostasis and dendritic spine shape by controlling ion and water homeostasis; (ii) the actin cytoskeleton dynamics through its binding to cofilin-1 and the regulation of RhoGTPase activity [18], which, in the case of KCC2-cofilin interaction, controls AMPA receptor insertion during long-term potentiation; and (iii) the modulation of synaptic function and neuronal excitability by interacting directly or indirectly (i.e., through KCC2) with the GluK2 subunit of Kainate receptors and with its auxiliary subunit Neto2, Task-3 (KCNK9) leak potassium channels and GABAB receptors (references in [4]). It is noteworthy that co-expression in the oocytes of NKCC1 and KCC4 doubles chloride import compared to oocytes expressing only NKCC1 with no change in NKCC1 expression level [43], suggesting that a cooperative interaction occurs between NKCC1 and KCC4 in oocytes. Therefore, one function of KCC2-NKCC1 co-clustering in mature neurons could be to fine-tune chloride fluxes but also synaptic activity and plasticity.

To be organized in membrane clusters, NKCC1 transporters are probably anchored to the actin cytoskeleton via their binding to scaffolding proteins. NKCC1 binds to the actin regulatory molecule cofilin-1 [18], and NKCC1 activity is regulated by the cytoskeleton (ref in [5]). However, to the best of our knowledge, no scaffolding proteins of NKCC1 have been identified to date. KCC2 binds to the actin cytoskeleton through the interaction of its carboxy-terminal domain with the 4.1N protein [44]. KCC2 diffusion constraints are relieved upon 4.1N suppression, the expression of the KCC2 carboxy-terminal domain or actin depolymerization with Latrunculin A, suggesting that a subpopulation of NKCC1 clusters may be anchored at/near synapses via KCC2-4.1N-actin binding [21]. Future studies will tell whether distinct scaffolding molecules anchor NKCC1a and NKCC1b in the dendrite vs. the AIS and axon. However, the binding of NKCC1 to the dendritic cytoskeleton is probably weaker than that anchoring KCC2 to the cytoskeleton since we show here using conventional microscopy and STORM that NKCC1a and NKCC1b clusters are smaller and significantly less dense in molecules than KCC2 clusters. These different clustering properties were associated with the fact that NKCC1a and NKCC1b are much more mobile than KCC2 on the dendritic surface.

We found that NKCC1a/b transporters are more confined to endocytic zones than on the dendrite. This differs from KCC2, whose mobility is equivalent inside and outside endocytic zones. This suggests that NKCC1a/b are more strongly anchored to the cytoskeleton of clathrin-coated pits than to the cytoskeleton of the dendritic plasma membrane. However, NKCC1a/b can escape from these endocytic zones because their dwell time there is much lower than at/near inhibitory synapses. Although this has little influence on the amount/function of NKCC1a/b on the dendritic surface under basal activity conditions, it may play a role in pathological situations associated with an increased expression of NKCC1 [45]. Endocytic zones have been shown to constitute reserve pools of molecules, which can, depending on the synaptic demand, be released and reintegrated into the diffusing pool of receptors [46]. In the case of NKCC1a/b, this reserve pool may allow a rapid increase in the transporter availability in the dendritic plasma membrane, e.g., in pathological situations in which the upregulation of NKCC1 has been observed [45]. In a recent work, we reported that exposure to the convulsive agent 4-aminopyridin increases NKCC1 membrane expression and clustering, leading to intracellular chloride influx [37]. In the pathology, the upregulation of NKCC1 is often accompanied by the downregulation of KCC2 on the neuronal surface [47,48]. KCC2 is also regulated by the diffusion–capture mechanism: 4-aminopyridin treatment increases the lateral diffusion of KCC2 transporters, which escape the clusters and are internalized and degraded [21]. Thus, lateral diffusion may be a general mechanism to control membrane stability and therefore the function of the chloride cotransporters NKCC1a/b and KCC2 (Figure 9). The inhibition of the pathway controlling the lateral diffusion of NKCC1a/b and KCC2 could block the increase in NKCC1a/b and the loss of KCC2 on the neuronal surface, thus potentially preventing the abnormal rise in [Cl^−^]_i_ in the pathology and the resulting adverse effects.

## Figures and Tables

**Figure 1 cells-12-02363-f001:**
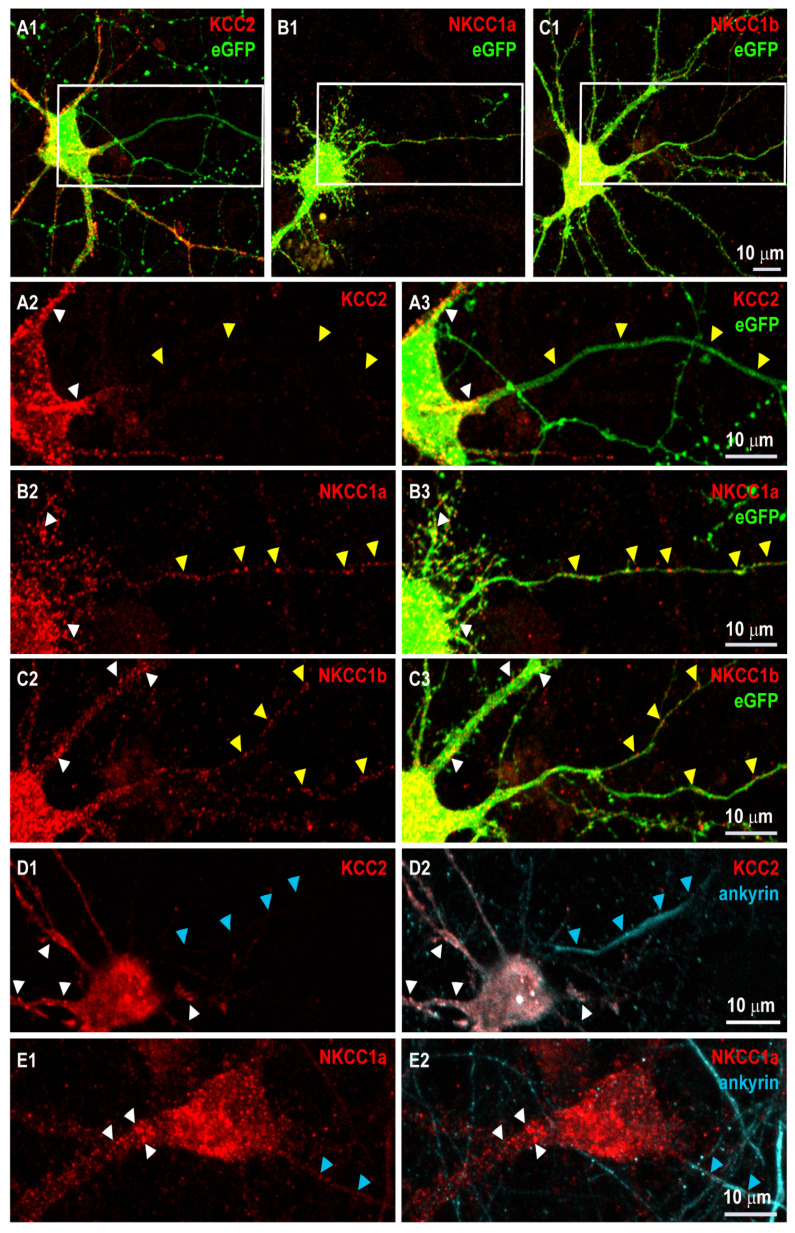
NKCC1a and NKCC1b but not KCC2 are present in the axon initial segment and along axons of mature hippocampal neurons. (**A1**–**C3**) Surface staining of KCC2 (red in **A1**–**A3**), NKCC1a (red in **B1**–**B3**), NKCC1b (red in **C1**–**C3**) in 21-DIV-old hippocampal neurons corresponding to the magnification of the dendritic regions outlined in (**A1**–**C1**), respectively with KCC2-Flag (**A1**–**A3**), NKCC1a-HA (**B1**–**B3**) or NKCC1b-HA (**C1**–**C3**) constructs and cytoplasmic eGFP (green in **A1**,**B1**,**C1**,**A3**,**B3**,**C3**). Panels (**A2**,**A3**,**B2**,**B3**,**C2**,**C3**). Scale bars: 10 µm. KCC2, NKCC1a and NKCC1b form clusters around somata and dendrites (white arrowheads in **A2**,**A3**,**B2**,**B3**,**C2**,**C3**,**D1**,**D2**,**E1**,**E2**, respectively). Note that KCC2 labeling is absent from the thin eGFP-positive axon-like process (yellow arrowheads in **A2**,**A3**), while NKCC1a and NKCC1b form clusters along similar neurites (yellow arrowheads in **B2**,**B3**,**C2**,**C3**, respectively). (**D1**–**E2**) Double-labeling of KCC2 (red in **D1**,**D2**) or NKCC1a (red in **E1**,**E2**) and ankyrin (blue in **D2**, **E2**, respectively) in mature neurons reporting NKCC1a clusters in the initial segment of the axon (blue arrowheads in **D1**,**D2**,**E1**,**E2**). Scale bars: 10 µm.

**Figure 2 cells-12-02363-f002:**
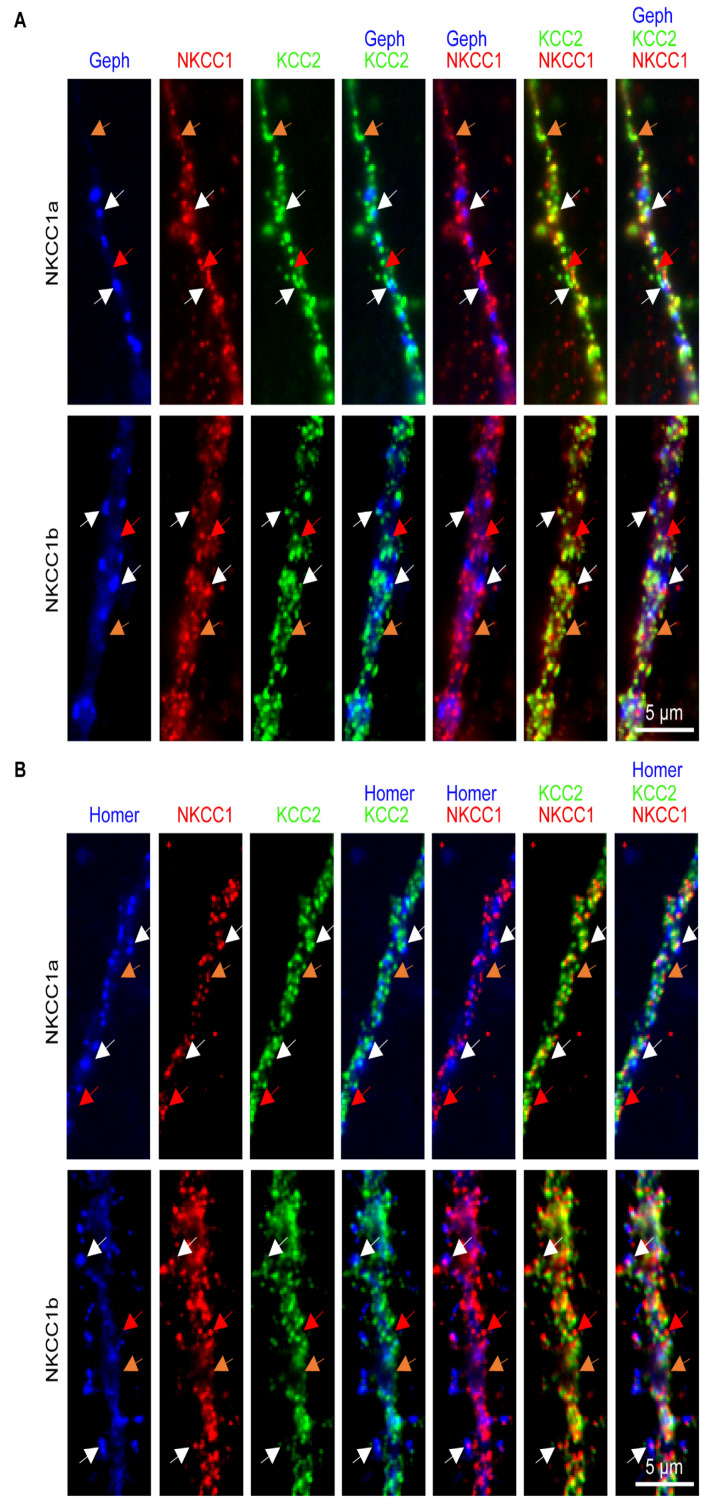
NKCC1a or NKCC1b colocalize with KCC2 at/near synapses. (**A**,**B**) Surface staining of KCC2 (green in **A**,**B**) and NKCC1a (red in **A**,**B**, upper row) or NKCC1b (red in **A**,**B**, lower row) in 21-DIV-old hippocampal neurons transfected with KCC2-Flag, NKCC1a-HA or NKCC1b-HA constructs and gephyrin-FingR-eGFP or homer1c-GFP to label GABAergic (blue in **A**) and glutamatergic (blue in **B**) synapses, respectively. Scale bars: 5 µm. Note that, here, we selected images where NKCC1a/b are expressed at a high level in order to better visualize their distribution compared to that of KCC2. Many dendritic NKCC1a or NKCC1b clusters colocalize with KCC2 clusters (white and orange arrows). Note that NKCC1a/1b-KCC2 clusters (white arrows) are bigger than NKCC1a or NKCC1b clusters devoid of KCC2 (red arrows) and that extrasynaptic NKCC1a/b-KCC2 clusters (orange arrows) are smaller than NKCC1a/b-KCC2 clusters at/near synapses (white arrows).

**Figure 3 cells-12-02363-f003:**
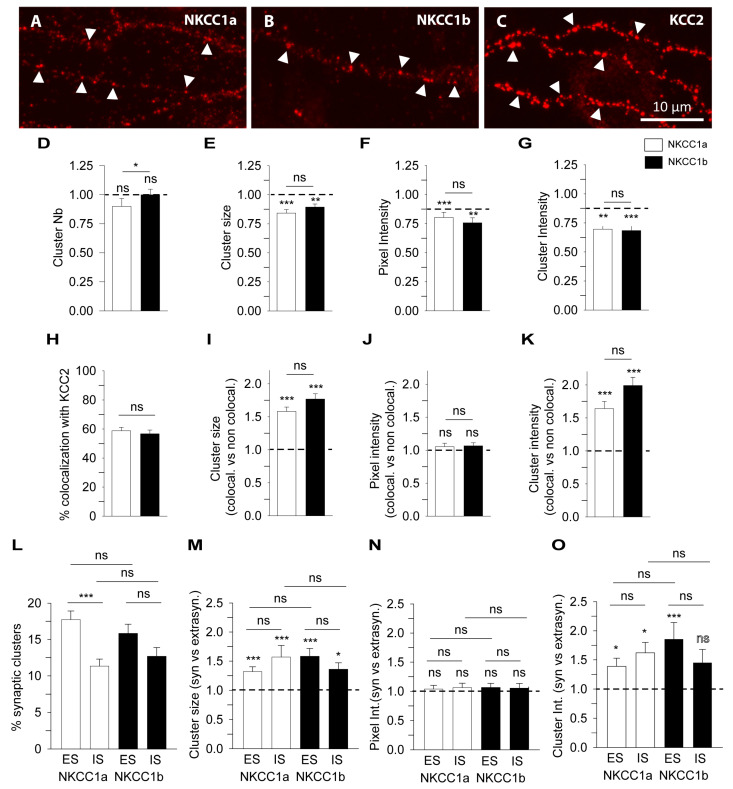
Quantitative analysis of NKCC1a and NKCC1b clustering using conventional widefield microscopy. (**A**–**C**) Surface staining of NKCC1a (**A**), NKCC1b (**B**) and KCC2 (**C**) in 21-DIV-old hippocampal neurons transfected with NKCC1a-HA, NKCC1b-HA or KCC2-Flag constructs. Scale bar: 10 µm. Note that NKCC1a, NKCC1b and KCC2 form many clusters along the dendrites (arrowheads) and that the intensity of KCC2 clusters is stronger than that of NKCC1a and NKCC1b clusters. (**D**–**G**) NKCC1a and NKCC1b clustering compared to KCC2 clustering. Quantifications of NKCC1a (white bars) and NKCC1b (black bars) cluster number (**D**), cluster area (**E**), cluster pixel intensity (**F**) and cluster intensity (**G**) showing reduced clustering of NKCC1a and NKCC1b compared to that of KCC2. Note that NKCC1a clustering is similar to NKCC1b clustering. Data shown as mean ± SEM. In all graphs, values were normalized to the corresponding KCC2 mean values. NKCC1a, n = 49 cells, NKCC1b, n = 57 cells, 3 cultures. Mann–Whitney rank sum test (MW test), NKCC1a, Cluster Nb *p* = 8.0 × 10^−2^, area *p* < 1.0 × 10^−3^, pixel intensity *p* < 1.0 × 10^−3^, cluster intensity *p* = 2.0 × 10^−3^. NKCC1b, Cluster Nb *p* = 0.72, area *p* = 3.0 × 10^−2^, pixel intensity *p* = 2.0 × 10^−3^, cluster intensity *p* < 1.0 × 10^−3^. NKCC1a vs. NKCC1b, Cluster Nb *p* = 2.8 × 10^−2^, area *p* = 0.14, pixel intensity *p* = 0.32, cluster intensity *p* = 0.65. (**H**–**K**) NKCC1a and NKCC1b colocalize with KCC2. (**H**) Percentage of NKCC1a (white bars) and NKCC1b (black bars) clusters colocalized with KCC2 clusters. Data shown as mean ± SEM. MW test, NKCC1a vs. NKCC1b, *p* = 0.65. (**I**–**K**) Quantifications of NKCC1a (white bars) and NKCC1b (black bars) cluster area (**I**), cluster pixel intensity (**J**) and cluster intensity (**K**) for NKCC1 clusters colocalized with KCC2 (NKCC1-KCC2) clusters. Note that NKCC1a/b-KCC2 clusters are bigger and more intense than the non-colocalized NKCC1a or NKCC1b clusters. Data shown as mean ± SEM. Values were normalized to the corresponding non-colocalized NKCC1a or NKCC1b mean values. MW test, NKCC1a, cluster area *p* < 1.0 × 10^−3^, pixel intensity *p* = 0.36, cluster intensity *p* < 1.0 × 10^−3^. NKCC1b, cluster area *p* < 1.0 × 10^−3^, pixel intensity *p* = 0.2, cluster intensity *p* < 1.0 × 10^−3^. NKCC1a vs. NKCC1b, cluster area *p* = 9.8 × 10^−2^, pixel intensity *p* = 1.00, cluster intensity *p* = 2.8 × 10^−2^. (**L**–**O**) Quantifications of NKCC1a and NKCC1b clustering at/near GABAergic and glutamatergic synapses. (**L**) Percentage of NKCC1a (white bars) and NKCC1b (black bars) clusters at/near excitatory synapses (ESs) and inhibitory synapses (ISs). Note that subtypes of NKCC1a and NKCC1b clusters are detected at/near excitatory and inhibitory synapses. Data shown as mean ± SEM. MW test, NKCC1a, n_SE_ = 32, n_SI_ = 36; NKCC1b, n_SE_ = 35, n_SI_ = 36; 2 cultures. MW test, NKCC1a, SE vs. SI *p* < 1.0 × 10^−3^; NKCC1b, SE vs. SI *p* = 7.8 × 10^−2^. SE, NKCC1a vs. NKCC1b *p* = 0.21. SI, NKCC1a vs. NKCC1b *p* = 0.61. (**M**–**O**) Quantifications of NKCC1a (white bars) and NKCC1b (black bars) cluster area (**M**), cluster pixel intensity (**N**) and cluster intensity (**O**) at excitatory (ES) and inhibitory (IS) synapses. Note that NKCC1a and NKCC1b clusters are bigger and more intense at excitatory and inhibitory synapses than those at the extrasynaptic ones. Data shown as mean ± SEM. Values were normalized to the corresponding extrasynaptic mean values. MW test, NKCC1a, ES: cluster area *p* < 1.0 × 10^−3^, pixel intensity *p* = 0.71, cluster intensity *p* = 0.01. NKCC1a, IS: cluster area *p* < 1.0 × 10^−3^, pixel intensity *p* = 0.71, cluster intensity *p* = 0.01. NKCC1a, ES vs. IS: cluster area *p* = 0.47, pixel intensity *p* = 0.76, cluster intensity *p* = 0.32. NKCC1b, ES: cluster area *p* < 1.0 × 10^−3^, pixel intensity *p* = 0.46, cluster intensity *p* < 1.0 × 10^−3^. NKCC1b, IS: cluster area *p* = 2.2 × 10^−2^, pixel intensity *p* = 0.59, cluster intensity *p* = 0.11. NKCC1b, ES vs. IS: cluster area *p* = 0.11, pixel intensity *p* = 0.80, cluster intensity *p* = 5.3 × 10^−2^. NKCC1a vs. NKCC1b, ES: cluster area *p* = 6.7 × 10^−2^, pixel intensity *p* = 0.85, cluster intensity *p* = 0.16. NKCC1a vs. NKCC1b, IS: cluster area *p* = 0.48, pixel intensity *p* = 0.86, cluster intensity *p* = 0.11. *, *p* < 5.0 × 10^−2^, **, *p* < 1.0 × 10^−2^, ***, *p* < 1.0 × 10^−3^ (Mann–Whitney rank sum test). ns, not significant.

**Figure 4 cells-12-02363-f004:**
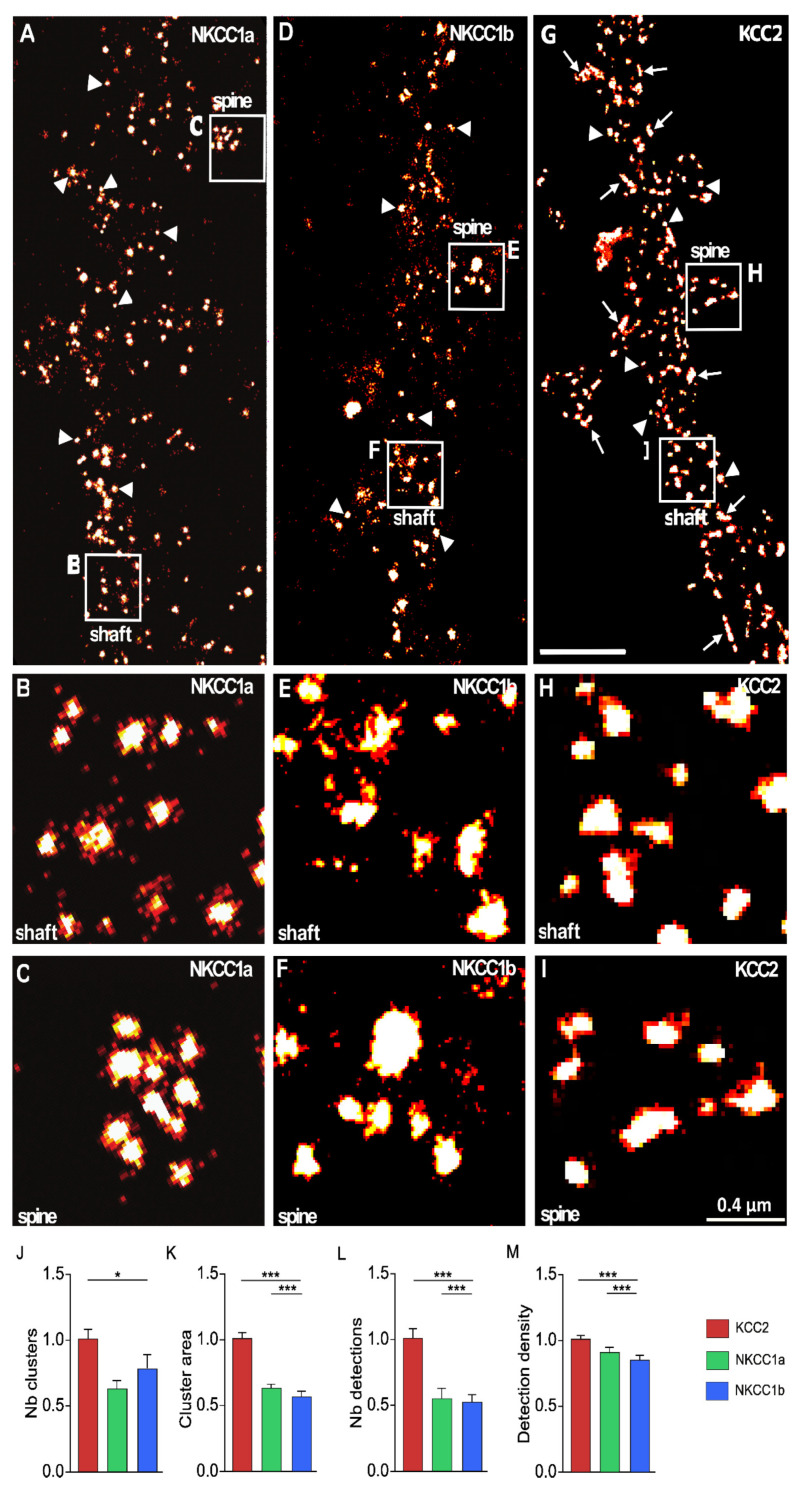
Nanoscale distributions of NKCC1a, NKCC1b and KCC2 transporters on the dendritic surface. (**A**–**I**) STORM images of membrane-associated NKCC1a (**A**–**C**), NKCC1b (**D**–**F**) and KCC2 (**G**–**I**) in hippocampal neurons at 21 DIV. Panels (**B**,**C**,**E**,**F**,**H**–**I**) correspond to the magnification of the shaft and spines outlined on the neurons in (**A**,**D**,**G**). Scale bars: 2 µm in (**A**,**D**,**G**); 0.4 µm in (**B**,**C**,**E**,**F**,**H**–**I**). Note that NKCC1a and KCC2 form many clusters round in shape (arrowheads) along the dendritic shaft and in spines. Some elongated (arrows) clusters of KCC2 are also observed. (**J**–**M**) Quantifications of NKCC1a (green bars), NKCC1b (blue bars) and KCC2 (red bars) cluster number per dendritic length (**J**), cluster area (**K**), number of single molecules per cluster (**L**) and density of molecules per cluster (**M**) show reduced NKCC1a and NKCC1b clustering as compared to KCC2 clustering. Data shown as mean ± SEM. NKCC1a n = 822 clusters (5 cells), NKCC1b n = 1124 clusters (13 cells), KCC2 n = 1450 clusters (7 cells), 2 cultures. (**J**): KCC2 vs. NKCC1a *p* = 0.13, KCC2 vs. NKCC1b *p* = 0.05, NKCC1a vs. NKCC1b *p* = 0.64. (**K**): KCC2 vs. NKCC1a *p* = 0.53, KCC2 vs. NKCC1b *p* < 1.0 × 10^−3^, NKCC1a vs. NKCC1b *p* < 1.0 × 10^−3^. (**L**): KCC2 vs. NKCC1a *p* = 0.38, KCC2 vs. NKCC1b *p* < 1.0 × 10^−3^, NKCC1a vs. NKCC1b *p* < 1.0 × 10^−3^. (**M**): KCC2 vs. NKCC1a *p* = 0.08, KCC2 vs. NKCC1b *p* < 1.0 × 10^−3^, NKCC1a vs. NKCC1b *p* < 1.0 × 10^−3^. *, *p* < 5.0 × 10^−2^; ***, *p* < 1.0 × 10^−3^ (Mann–Whitney rank sum test).

**Figure 5 cells-12-02363-f005:**
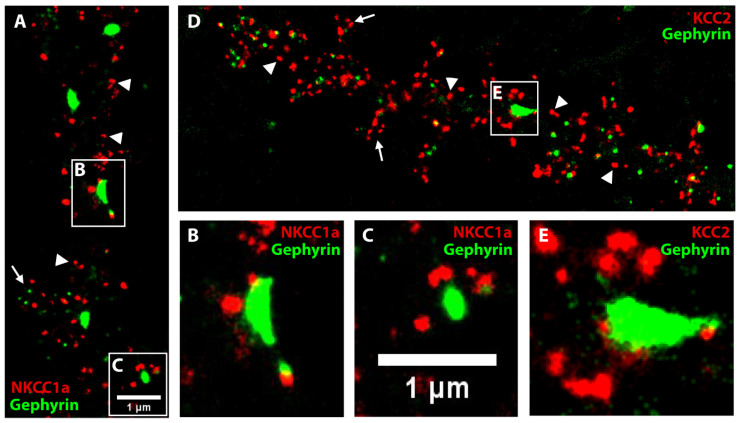
NKCC1a and KCC2 clusters surround inhibitory synapses. (**A**–**E**) Two-color STORM of NKCC1a (red in **A**–**C**) or KCC2 (red in **D**–**E**) and PALM of gephyrin (green in **A**–**E**) in 21-DIV-old neurons. Panels (**B**,**C**,**E**) correspond to the magnification of the synapses outlined on the neurons in **A**,**D**. Scale bars: 1 µm. Note that some NKCC1a and KCC2 clusters surround inhibitory synapses formed on the dendritic shaft. NKCC1a and KCC2 clusters are also detected distant to inhibitory synapses on the dendritic shaft (arrowheads in **A**,**D**, respectively) and in the spines (arrows in **A**,**D**, respectively).

**Figure 6 cells-12-02363-f006:**
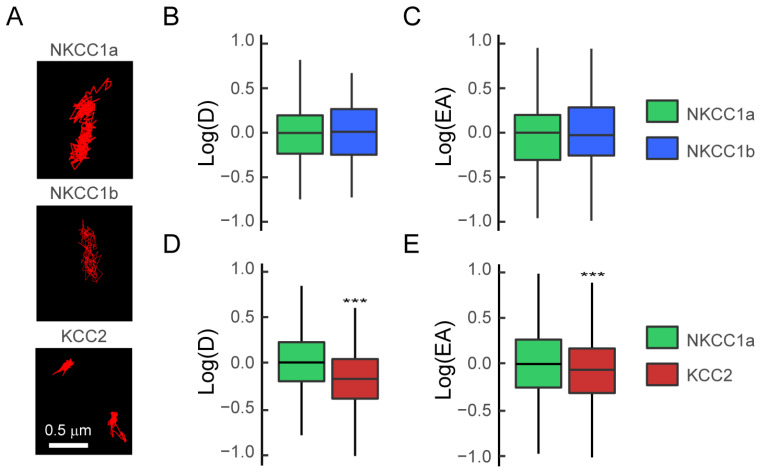
NKCC1a and NKCC1b are more mobile than KCC2 on the dendritic surface. (**A**) Representative trajectories (red) of NKCC1a, NKCC1b and KCC2 on the surface of dendrites of mature 21-DIV-old neurons. Scale bar: 0.5 µm. (**B**,**C**) Similar log(D) (**B**) and log(EA) (**C**) for the bulk (extrasynaptic + synaptic) population of NKCC1a (green) and NKCC1b (blue). Diffusion coefficient (**D**): NKCC1a n = 146 QDs, NKCC1b n = 116 QDs, *p* = 0.19, 1 culture. Explored area (EA): *p* = 0.63. (**D**,**E**) Reduced log(D) (D) and log(EA) (**E**) for the bulk population of KCC2 (red) as compared to NKCC1a (green). Diffusion coefficient (**D**): NKCC1a n = 180 QDs, KCC2 n = 675 QDs, *p* < 2.2 × 10^−16^, 3 cultures. Explored area (EA): *p* < 2.2 × 10^−16^. In (**B**–**E**), data are presented as median values ± 25%–75% IQR. Values in (**B**,**C**–**E**) were normalized to the corresponding NKCC1a values. ***, *p* < 1.0 × 10^−3^ (Welch *t*-test).

**Figure 7 cells-12-02363-f007:**
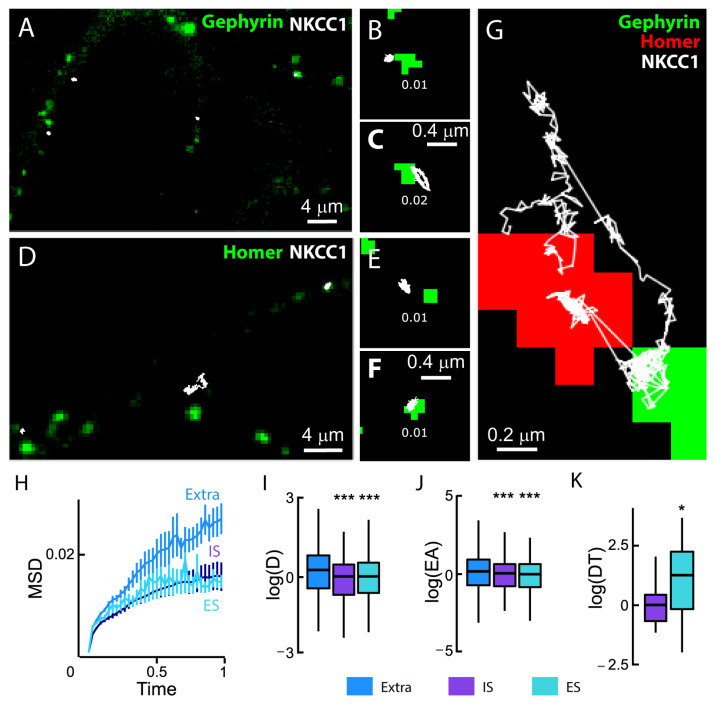
NKCC1a diffusion at synaptic and extrasynaptic sites. (**A**–**F**) Representative trajectories (white) of QD-bound HA-tagged NKCC1a on the surface of neurons transfected with gephyrin-FingR-eGFP (**A**–**C**) or homer1c-DsRed (**D**–**F**). QD trajectories (white) were overlaid with fluorescent clusters of gephyrin-FingR-eGFP (green in **A**–**C**) and Homer1c-DsRed (green in **D**–**F**) to identify inhibitory (IS) and excitatory (ES) synapses, respectively. Mean diffusion coefficient values (µm^2^·s^−1^) are displayed for each trajectory. Scale bars: 4 µm in A and D and 0.4 µm in (**B**,**C**,**E**,**F**,**G**). An individual NKCC1a transporter shifts between an excitatory (green) and an inhibitory synapse (red). Scale bar: 0.2 µm. (**H**) Time-averaged MSD vs. time functions of extrasynaptic QDs (blue), QDs at/near excitatory synapses (turquoise) and QDs at/near inhibitory synapses (purple). The MSD (µm^2^) versus time (s) relationship for extrasynaptic trajectories shows a steeper initial slope, suggesting that trajectories were less confined. Extra, n = 747 QDs, IS, n = 209 QDs, ES, n = 369 QDs, 5 cultures. (**I**,**J**) Boxplots of log(D) (**I**) and log(EA) (**J**) showing a reduced mobility and an increased confinement of NKCC1a at/near IS (purple) and ES (turquoise) compared to transporters distant to synapses (blue). Welch *t*-test, diffusion coefficient (**D**): IS, *p* = 1.58 × 10^−14^; ES, *p* = 8.4 × 10^−13^. Explored area (EA): IS, *p* = 5.6 × 10^−11^; ES, *p* = 7.2 × 10^−10^. (**K**) Boxplots of log(dwell time, DT) of NKCC1a at/near inhibitory (purple) and excitatory (turquoise) synapses showing increased DT at/near excitatory synapses as compared with inhibitory synapses. MW test, *p* = 0.03. *, *p* < 5.0 × 10^−2^, ***, *p* < 1.0 × 10^−3^ (Welch *t*-test).

**Figure 8 cells-12-02363-f008:**
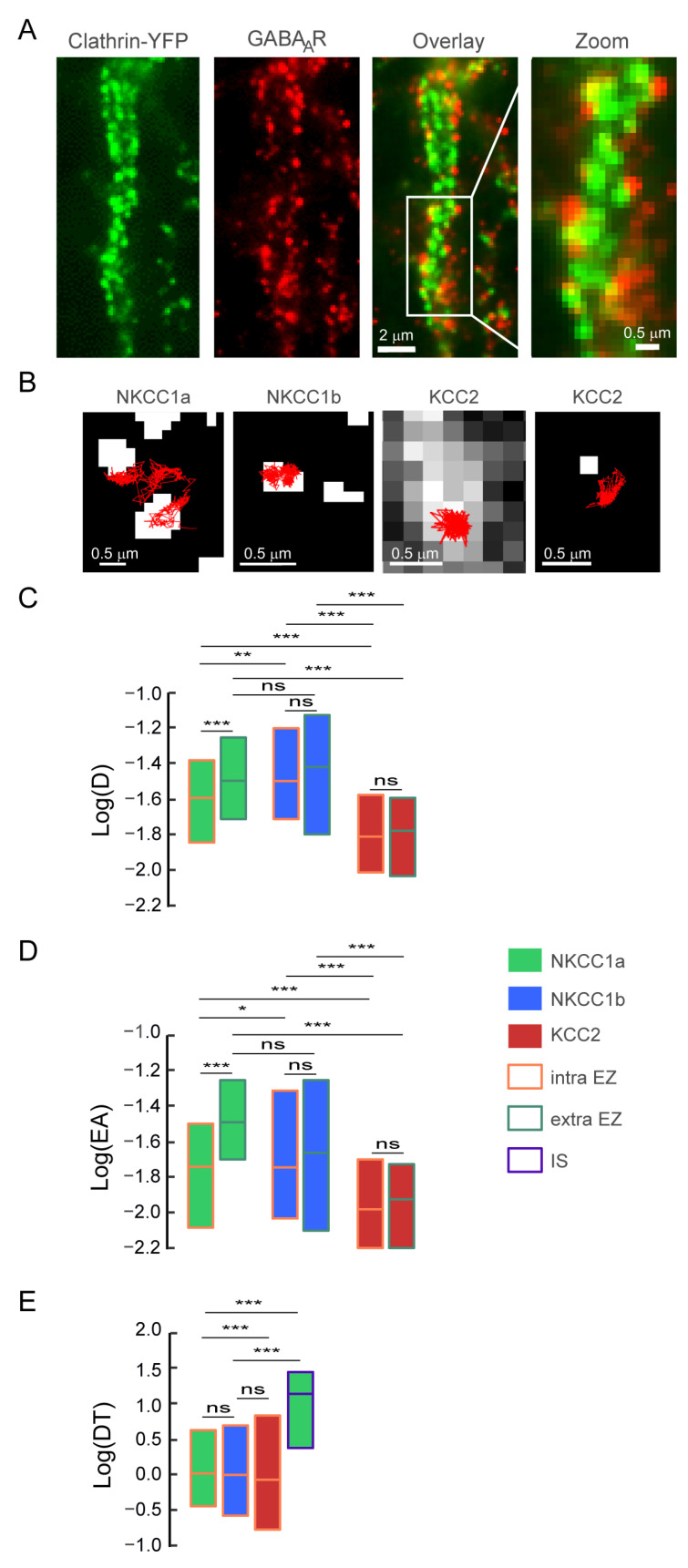
Confinement of NKCC1a/1b within endocytic zones. (**A**) Representative images of clathrin-YFP (green) and GABA_A_R γ2 subunit (red) in neurons transfected at DIV14 with clathrin-YFP and stained at DIV21 for the GABA_A_R γ2 subunit. Overlay shows that GABA_A_Rγ2 clusters are surrounded by clathrin-YFP clusters. Scale bars: 2 µm and 0.5 µm as indicated. (**B**) Representative trajectories (red) of NKCC1a, NKCC1b and KCC2 in relation to endocytic zones (white) identified by the presence of clathrin-YFP clusters. Scale bars: 0.5 µm. (**C**,**D**) Quantifications of log(D) (**C**) and log(EA) (**D**) for NKCC1a (green bars), NKCC1b (blue bars) and KCC2 (red bars) inside (orange lines) endocytic zones (EZs) as compared to QDs outside (green lines) EZs. NKCC1a are slower and more confined than NKCC1b and KCC2 inside vs. outside EZ. Note also that KCC2 diffusion properties are similar inside and outside EZ. C: NKCC1a, intra-EZ n = 210 QDs, extra-EZ n = 401 QDs; NKCC1b, intra-EZ n = 115 QDs, extra-EZ n = 305 QDs; KCC2, intra-EZ n = 250 QDs, extra-EZ n = 555 QDs, 3 cultures. MW test, NKCC1a, intra-EZ vs. extra-EZ *p* < 1.0 × 10^−3^; NKCC1b, intra-EZ vs. extra-EZ *p* = 0.76; KCC2, intra-EZ vs. extra-EZ *p* = 0.75. Intra-EZ, NKCC1a vs. NKCC1b *p* = 4.0 × 10^−3^, NKCC1a vs. KCC2 *p* < 1.0 × 10^−3^, NKCC1b vs. KCC2 *p* < 1.0 × 10^−3^. Extra-EZ, NKCC1a vs. NKCC1b *p* = 0.29, NKCC1a vs. KCC2 *p* < 1.0 × 10^−3^, NKCC1b vs. KCC2 *p* < 1.0 × 10^−3^. D: NKCC1a, intra-EZ n = 630 QDs, extra-EZ n = 1203 QDs; NKCC1b, intra-EZ n = 344 QDs, extra-EZ n = 913 QDs; KCC2, intra-EZ n = 723 QDs, extra-EZ n = 1575 QDs, 3 cultures. NKCC1a, intra-EZ vs. extra-EZ *p* < 1.0 × 10^−3^; NKCC1b, intra-EZ vs. extra-EZ *p* = 0.67; KCC2, intra-EZ vs. extra-EZ *p* = 0.11. Intra-EZ, NKCC1a vs. NKCC1b *p*= 4.0 × 10^−2^, NKCC1a vs. KCC2 *p* < 1.0 × 10^−3^, NKCC1b vs. KCC2 *p* < 1.0 × 10^−3^. Extra-EZ, NKCC1a vs. NKCC1b *p* = 0.88, NKCC1a vs. KCC2 *p* < 1.0 × 10^−3^, NKCC1b vs. KCC2 *p* < 1.0 × 10^−3^. (**E**) Boxplots of log(dwell time, DT) of NKCC1a (green bars), NKCC1b (blue bars) and KCC2 (red bars) in EZ (orange lines). Note that NKCC1a and NKCC1b spend more time in EZ than KCC2. NKCC1a, n_EZ_ = 196 QDs, NKCC1b, n_EZ_ = 190 QDs, KCC2, n_EZ_ = 450 QDs, NKCC1a, n_SI_ = 170 QDs, 3 cultures. MW test, intra-EZ, NKCC1a vs. NKCC1b *p* = 0.94; NKCC1a vs. KCC2 *p* < 1.0 × 10^−3^; NKCC1b vs. KCC2 *p* = 0.27. NKCC1a intra-EZ vs. IS *p* < 1.0 × 10^−3^. In all graphs, data are presented as median values ± 25–75% IQR. In (**C**,**D**), values were not normalized. In (**E**), values were normalized to NKCC1a intra-EZ. ns, not significant, *, *p* < 5.0 × 10^−2^, **, *p* < 1.0 × 10^−2^, ***, *p* < 1.0 × 10^−3^ (Mann–Whitney rank sum test).

**Figure 9 cells-12-02363-f009:**
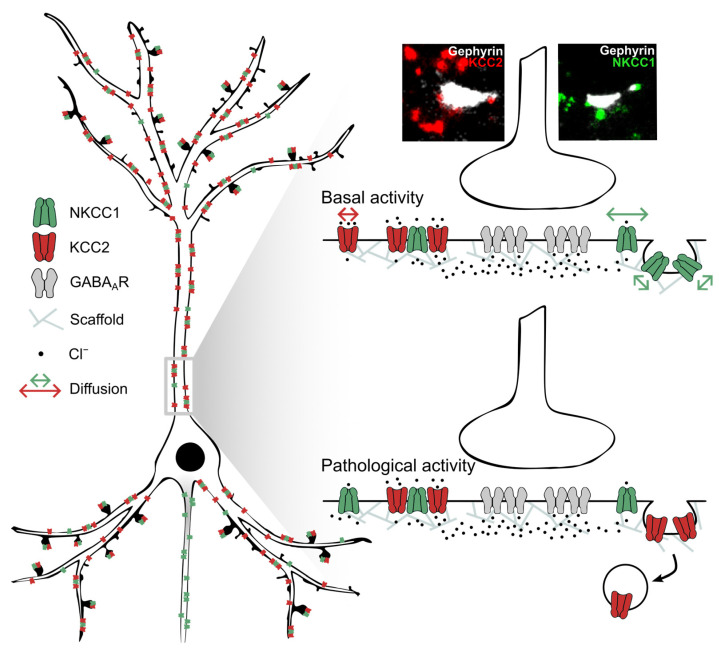
Opposing effects of pathological activity on the membrane diffusion and clustering of NKCC1 and KCC2. (**Left**) Note the low membrane clustering of NKCC1a/b near synapses and at extrasynaptic sites compared to KCC2 and the axonal targeting of NKCC1 but not KCC2. Most NKCC1a/b are co-clustered with KCC2. (**Right**) Under basal activity conditions (**top**), NKCC1 diffuses faster in the plasma membrane than KCC2, while it is more confined to clathrin-coated pits, suggesting that it is stored in endocytic zones. Under conditions of pathological activity (4-amynopyridin treatment), NKCC1 can escape endocytic zones to increase the membrane pool [37], while KCC2 escapes the membrane clusters to be internalized [21].

## Data Availability

The data that support the findings of this study are available from the corresponding author upon reasonable request. The transfer of plasmids generated for this study will be made available upon request. A Materials Transfer Agreement may be required.

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
