# Peer review of "NKCC1 and KCC2 Chloride Transporters Have Different Membrane Dynamics on the Surface of Hippocampal Neurons"

_cells, 2023, doi:10.3390/cells12192363_

Round 1
Reviewer 1 Report
The manuscript provides a quantitative analysis of localisation and dynamics of exogenously expressed Na+-K+ -Cl- co-transporter, NKCC1, clusters in cultured hippocampal neurons using fluorescent labelling and high resolution microscopy. NKCC1 and the related K+-Cl- cotransporter (KCC2) together play an important role in intracellular neuronal Cl- levels and homeostasis, thereby influencing the efficacy of inhibitory synaptic signalling and cell excitability. This study extends on work from the group that has characterised the mobility and localisation of KCC2, and a strength of the study is how the same parameters are now directly compared between two NKCC1 isoforms (a and b) and KCC2. This is important because the parameters for defining cluster and colocalization parameters can be somewhat arbitrary, but having the comparisons between the NKCC1s and KCC2 in the same samples with the same parameters allows more robust conclusions.
In my opinion the study provides novel data showing that: 1) NKCC1a/1b can be co-localised around both excitatory and inhibitory synapses, often with KCC2; 2) KCC2 has different membrane diffusion and dynamics (slower) than NKCC1 a/b and 3) AIS localisation of NKCC1 is interesting, although this is not quantified across a larger sample.
The methods and data is clearly displayed and described.
An interesting study, well done. I have a few points below.
1) Lines 361-368. I’m not sure the extent that different technical conditions (e.g., construct expression efficiency) contribute to the different detected levels of KCC2 and NKCC1a/b protein expression. The last sentence is interesting but is not valid without some direct testing levels. This needs to be dropped. You could state the values but I wouldn’t draw too much conclusions and may consider leaving the whole paragraph out (depending on how confident you feel about the relative values quoted).
2) Lines 449-452. Similarly, I would suspect the fluorescent intensity comparisons between KCC2 and NKCC1 may be subject to different technical conditions (eg different fluorophore environment when tagged to KCC2 vs NKCC1). Are these interpretations around the differential intensity being due to different numbers of molecules within a cluster valid? I feel a sentence is needed to state why these are so directly related or a sentence of caution about alternative interpretations. Same for lines 507-508.
3) Figure 3. Panels H-K. I think a Y axis label that distinguishes these from D-G would be helpful (ie indicate somehow these are KCC2 co-localised clusters). And can the KCC2 data also be indicated on panel L – as dashed lines or through another set of columns.
4) I wasn’t clear if NKCC1a and NKCC1b can be in the same cluster. Or are they distinct clusters? I think this is an interesting point.
5) If possible, quantify samples and frequency of axons/cells/cultures where NKCC1 was observed in Ankryin identified AIS as compared to KCC2 clusters in these compartments.
6) Lines 840-842 is nice conclusion but just tone down a bit, ie “could block”, “potentially preventing”.
Line 44. Define EGABA
Line 51. “NKCC1-mediated depolarization of GABAARs” rephrase. It’s a GABA-mediated depolarization enabled by NKCC1 Cl influx.
Line 57-60. “In the context of long-term potentiation, the interaction of NKCC1 with cofilin would regulate the dynamics of astrocytic processes via combined actions on the actin network and water influx, which would allow glutamate spillover and long-term potentiation [16].” I don’t really follow this – rephrase more clearly.
Line 67 “in the plasma membrane or as clusters located for many at the periphery” rephrase to indicate you mean that many of the clusters are located around GABA synapses as defined by. XX – maybe two sentences.
Line 132. Define “SPT” when 1st used.
Line 210 or later in results. Define what you mean by an “endocytic zone”
Line 234. “synaptic when overlapping with the synaptic mask of GPHN.FingR-eGFP and homer1c- DsRed, or extrasynaptic for spots two pixels (380 nm) away” Defining what these flurophores target would help. And don’t use “synaptic mask”., rewrite – e.g., “to identify inhibitory and excitatory synapses we used Synaptic or extrasynaptic location was defined as overlapping fluorescence or flurescence separated by at least two pixels (340 nm).”
Line 248. A symbol is missing, presumably “<”
Line 311. Rephrase “signs of suffering” (e.g. reduced cell viability)
Lines 537 onwards is italicised? An error I think.
Lines 710. Clarify or rewrite more clearly “Therefore, NKCC1a can escape faster endocytic zones than inhibitory synapses.”
Line781 “ovocyte” - oocyte
Author Response
"Please see the attachment."

Reviewer 2 Report
By using a well characterized neuronal model i.e. hippocampal neurons in culture, Pol et al. have studied the localization and membrane dynamics of NKCC1a/1b and KCC2, two chloride cotransporters, the first importing chloride into neuron and the second one exporting it from neuron. To answer that question, they used STORM, STORM/PALM microscopy and overexpressed tagged-cotransporters either alone or in conjunction with differents tagged proteins allowing the visualisation of two subcellular compartments, the synapses and the endocytic zones.
They conclude that NKCC1a/1b have different membrane dynamics and clustering than KCC2 which helps to explain their lower level at the membrane while allowing rapid increase in the membrane under pathological conditions.
The research group has an excellent expertise in cell biology and microscopic analysis and has an excellent practice in culture of hippocampal neurons.
The team has published numerous excellent papers on the cell biology of GABAergic transmission and its regulation.
The experiments described in the manuscript are well performed and analyzed. The conclusion is appropriate. The manuscript is also clearly written.
Author Response
"Please see the attachment."
